# Endothelium-specific depletion of LRP1 improves glucose homeostasis through inducing osteocalcin

Hua Mao[1,2], Luge Li[1,2], Qiying Fan[1,2], Aude Angelini [1,2], Pradip K. Saha[3], Cristian Coarfa [4,5], Kimal Rajapakshe[4,5], Dimuthu Perera[4,5], Jizhong Cheng[6], Huaizhu Wu[1,2], Christie M. Ballantyne [1,2], Zheng Sun [3,4], Liang Xie[1,2] & Xinchun Pi [1,2✉]

The vascular endothelium is present within metabolic organs and actively regulates energy metabolism. Here we show osteocalcin, recognized as a bone-secreted metabolic hormone, is expressed in mouse primary endothelial cells isolated from heart, lung and liver. In human osteocalcin promoter-driven green fluorescent protein transgenic mice, green fluorescent protein signals are enriched in endothelial cells lining aorta, small vessels and capillaries and abundant in aorta, skeletal muscle and eye of adult mice. The depletion of lipoprotein receptor-related protein 1 induces osteocalcin through a Forkhead box O -dependent pathway in endothelial cells. Whereas depletion of osteocalcin abolishes the glucose-lowering effect of low-density lipoprotein receptor-related protein 1 depletion, osteocalcin treatment normalizes hyperglycemia in multiple mouse models. Mechanistically, osteocalcin receptor-G protein-coupled receptor family C group 6 member A and insulin-like-growth-factor-1 receptor are in the same complex with osteocalcin and required for osteocalcin-promoted insulin signaling pathway. Therefore, our results reveal an endocrine/paracrine role of endothelial cells in regulating insulin sensitivity, which may have therapeutic implications in treating diabetes and insulin resistance through manipulating vascular endothelium.

[1] Department of Medicine, Section of Athero & Lipo, Baylor College of Medicine, Houston, TX, USA. [2] Cardiovascular Research Institute, Baylor College of Medicine, Houston, TX, USA. [3] Department of Medicine, Division of Diabetes, Endocrinology & Metabolism, Diabetes Research Center, Baylor College of Medicine, Houston, TX, USA. [4] Departments of Molecular and Cellular Biology, Baylor College of Medicine, Houston, TX, USA. [5] Dan L. Duncan Cancer Center, Baylor College of Medicine, Houston, TX, USA. [6] Department of Medicine, Section of Nephrology, Selzman Institute for Kidney Health, Baylor College of Medicine, Houston, TX, USA. ✉email: xpi@bcm.edu

An estimated 30.3 million Americans (9.4% of the U.S. population) have diabetes and another 84.1 million people have pre-diabetes, a condition that frequently leads to type 2 diabetes mellitus (T2DM) within 5 years[1]. Individuals with T2DM, constituting about 90% of all cases of diabetes, are unable to properly use their insulin supply due to insulin resistance. Patients with type 1 diabetes mellitus (T1DM) rely on exogenous insulin for blood control and survival. Both T1DM and T2DM are accompanied with chronic vascular complications, which impose profound impacts on the quality of life and health care resource. A growing list of stimuli, including genetic mutations, lipotoxicity, glucotoxicity, inflammation, mitochondrial dysfunction and ER stress can trigger insulin resistance. However, insulin resistance pathogenesis is complicated. When lifestyle change and monotherapies (i.e., metformin, sulfonylurea or insulin) fail to keep hemoglobin A1C values low, combined therapy is needed to sustain glucose at the normal level[2]. However, these therapies are associated with high costs, variable effectiveness and a variety of side effects. It raises urgent needs to identify new molecules that enhance insulin signaling or can be used as insulin replacement for treating diabetes.

The crosstalk of metabolic organs plays a crucial role in glucose and lipid homeostasis, and its dysregulation contributes to the progression of diabetes and insulin resistance[3]. The close anatomical association between metabolic tissues and vascular endothelium suggests they may possess bidirectional crosstalk and thereby be functionally interdependent. However, the importance of endothelial dysregulation in the development of insulin resistance and diabetes is still not fully understood. Low-density lipoprotein receptor-related protein 1 (LRP1), a multifunctional member of the LDL receptor family, is involved in a variety of biological processes such as lipid metabolism, endocytosis and signal transduction[4]. Mice with global LRP1 depletion are embryonic lethal, however, tissue-specific knockout studies link hepatic, adipose, pancreatic and neuronal LRP1 to lipid metabolism, glucose homeostasis and obesity[5–10]. With studies of endothelial-specific LRP1 knockout mice, we have identified a pivotal role for LRP1 in angiogenesis and vascular inflammation[11–13]. We also discovered that endothelial LRP1 depletion improved systemic metabolic homeostasis through increasing lipid metabolism[14]. However, it is not fully understood how endothelial LRP1 regulates glucose homeostasis.

In this study, we performed RNA sequencing analysis with ECs isolated from LRP1 EC-specific inducible knockout (eKO) mice and identified osteocalcin (OCN, also called OCN1, OG1 or Bglap) as one of most upregulated genes in LRP1-depleted ECs. OCN has been recognized as an osteoblast-secreted metabolic hormone[15–18]. Interestingly, our studies demonstrate EC-LRP1 depletion increases OCN expression in ECs and its serum level in vivo. By using a human *ocn* promoter-driven GFPtpz reporter mouse model[15], GFP signals are enriched in ECs lining aorta, small vessels and capillaries during adulthood. In addition, our results suggest EC-LRP1 depletion alleviates hyperglycemia and insulin resistance in diabetic mice through OCN. Taken for all, our data support that vascular endothelium exerts endocrine/paracrine regulation of energy homeostasis and expands the biological importance of this organ in glucose homeostasis.

## Results

**OCN is expressed in vascular endothelium and induced by LRP1 depletion**. Our previous studies indicate that endothelial LRP1 is involved in whole-body energy homeostasis by using an EC-specific LRP1 knockout mouse model generated by the cross of LRP1$^{f/f}$ and Tie2Cre$^{+}$ (LRP1$_{Tie2}$$^{-/-}$) mice followed by bone marrow transplantation[14]. Although LRP1-dependent regulation

of PPAR activity and metabolic gene induction explains how LRP1 depletion improves lipid homeostasis[14], it remains unclear how glucose homeostasis is improved by EC-depletion. Considering anatomic proximity between vascular endothelium and metabolic cells/tissues, we speculated that EC-LRP1 depletion might promote glucose metabolism through secreting regulators. Therefore, we employed mRNA-sequencing (mRNA-seq) analysis to screen ECs isolated from an EC-specific LRP1 inducible knockout mouse model (LRP1 eKO, LRP1$^{f/f}$; Cdh5-CreER$^{+/-}$, Supplementary Fig. 1a, b) and their littermate control (WT, LRP1$^{f/f}$; Cdh5-CreER$^{-/-}$) mice for potential EC-secreted factors. At a fold-change cutoff of $> +/-$ 2.0, there were sets of genes either upregulated (905, 1096) or downregulated (446, 257) in response to LRP1 depletion in ECs isolated from liver (MLivECs) or heart and lung (MHLECs), respectively (Fig. 1a, b and Supplementary Fig. 2a, b). Notably, gene expression patterns in MLivECs and MHLECs were overly different (Fig. 1b). Gene set enrichment analysis (GSEA) was performed with the MSigDB 6.0 on these datasets of MLivECs and MHLECs, and GSEA implementation was used to screen for pathways and processes. GSEA of data indicated that EC-LRP1 depletion resulted in changes of multiple cellular processes, such as proliferation, inflammation and metabolic responses, and including both overlapping and unique ones in MLivECs and MHLECs (Supplementary Fig. 2c). It suggests the heterogeneity of ECs isolated from different vessel beds. More importantly, both paracrine and endocrine signals may play roles in endothelial LRP1-dependent metabolic regulation.

*Ocn* was one among the most upregulated genes that have been identified in our mRNA-seq datasets, showing a 58.13- or 30.14-fold increase in LRP1-depleted MLivECs or MHLECs, respectively (Fig. 1a and Supplementary Table 1). To further understand how OCN expression is regulated in ECs, we first confirmed the induction of OCN and changes of other candidate genes with real-time PCRs (Supplementary Fig. 2d). OCN mRNA level was markedly increased up to 7.6- or 6.7-fold in LRP1-depleted MLivECs or MHLECs (Fig. 1c). As OCN is a secreted protein, we detected an increase in its serum level in LRP1 eKO mice (Fig. 1d). OCN can be carboxylated at γ-carboxyglutamic acid (Gla) residues, and only uncarboxylated (Glu) and undercarboxylated OCN are biologically active[17]. We noticed serum levels of both active and inactive OCN (Glu-OCN, Gla-OCN) were increased (Fig. 1d). Different from human *ocn* gene, there is a gene cluster containing *ocn1* (*ocn*), *ocn2* and *ocn*-related gene (*org*) in mouse genome[16]. OCN1 and OCN2 proteins only differ in two amino acids located at their signal peptides, while ORG is more different from them. We observed upregulation of both OCN1 and OCN2 in mRNA-seq data (Supplementary Table 1) and confirmed their increases in LRP1-depleted mouse lung ECs (MLECs, Fig. 1e) and MHLECs (Supplementary Fig. 2e–g). In addition, OCN protein was detected in conditioned media (CM) of human and mouse primary ECs and LRP1 depletion in MLECs by its specific siRNA dramatically increased its level (Fig. 1f). However, LRP1 depletion in ECs did not increase OCN levels in marrow-flushed bones or cultured osteoblasts and their CM isolated from LRP1 eKO mice (Fig. 1g). The specific increase of OCN level by LRP1 depletion in ECs but not in osteoblasts was also observed in their CM (Fig. 1h and Supplementary Fig. 2h). To test whether LRP1 depletion in ECs might impact bone remodeling that indirectly affects the circulating osteocalcin level, we evaluated bone turnover and observed no significant differences in the bone formation rates between LRP1 eKO and WT mice (Supplementary Fig. 2i). Taken together, our results suggest that EC-LRP1 depletion specifically increases OCN expression in ECs.

OCN has been recognized as an osteoblast-specific hormone[15–18]. However, its expression in ECs has never been

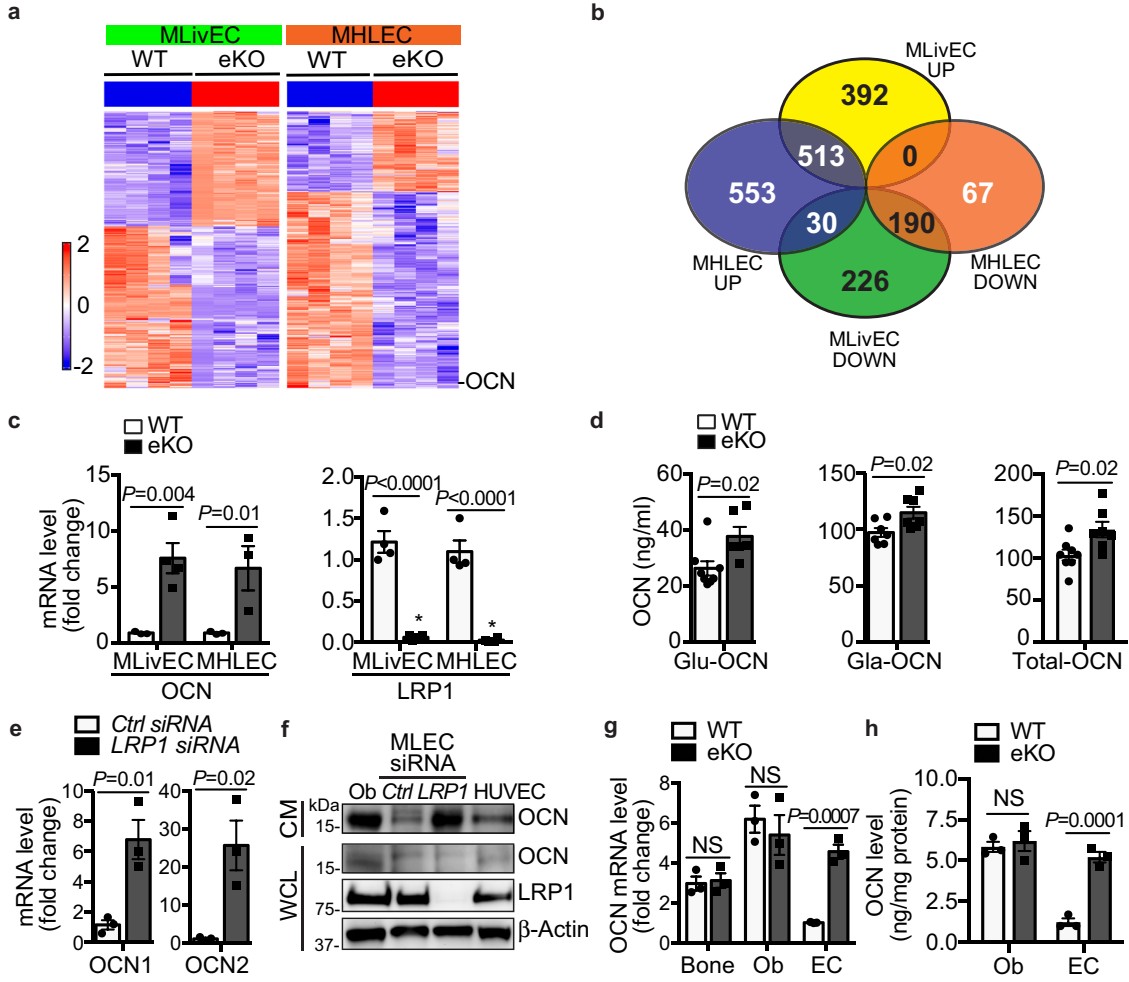

**Fig. 1 Osteocalcin is induced in LRP1-depleted ECs. a** Heatmaps of mRNA-seq data demonstrate changes in gene expression profiles of liver ECs (MLivECs) and heart and lung ECs (MHLECs) isolated from LRP1 (eKO, LRP1$^{f/f}$; Cdh5-CreER$^{+/-}$) or their littermate control (WT, LRP1$^{f/f}$; Cdh5-CreER$^{-/-}$) mice. **b** A Venn diagram shows upregulated (UP) or downregulated (DOWN) gene numbers in LRP1-depleted MLivECs and MHLECs. **c** Expression changes of OCN and LRP1 were confirmed by real-time PCR. **d** Blood levels of Glu-, Gla- and total-OCN. **e** OCN1 and OCN2 mRNA levels in mouse lung ECs (MLECs) following transfection of LRP1 or control siRNAs. **f** OCN levels in conditioned media (CM) and whole-cell lysates (WCL) of osteoblasts (Ob) and different ECs. HUVEC, human umbilical vein endothelial cell. **g** OCN mRNA levels in marrow-flushed bone, osteoblasts (Ob) and ECs isolated from wildtype (WT) or LRP1 eKO mice. **h** OCN levels in CM of Obs and ECs isolated from WT or LRP1 eKO mice. n = 3 (**c** WT; **c** eKO, MHLEC), 4 (**c** eKO, MLivEC), 8 (**d** WT, Glu/Total-OCN), 7 (**d** Gla-OCN; eKO, Total-OCN), 6 (**d** eKO, Glu-OCN), 3 (**e, g, h**). NS, not significant. Data are presented as mean ± SEM. Analysis was two-way ANOVA followed by Fisher's LSD multiple comparison test (**c, g, h**) or unpaired two-tailed Student's *t*-test (**d, e**).

reported. Previous studies suggest the promoters of the two mouse *ocn* genes exhibit the similar modular organization as that in the rat and human genes although some *cis*- and *trans*-acting elements and their responses to vitamin D are different[19–21]. Our data showed LRP1 depletion increased mouse and human OCN expression similarly (Fig. 1f), suggesting the regulation of OCN expression by LRP1 in mouse and human likely shares a common mechanism. Therefore, we evaluated OCN expression with hOC-GFPtpz transgenic mouse model, where GFP expression is driven by human *ocn* promoter[15]. Surprisingly, aorta, skeletal muscle and eye displayed strong GFP signals although their levels were lower than marrow-flushed bones (Fig. 2a). The *en face* and cross-section staining of aorta isolated from hOC-GFPtpz mice demonstrated these GFP signals were mainly located in ECs (Fig. 2b, top panel and 2c, Supplementary Fig. 2j). In addition, GFP signals were observed in ECs lining small vessels and capillaries of the skeletal muscle (Fig. 2b, bottom panels). We also monitored GFP signals in mice at different age and noticed a decrease of GFP signals in marrow-flushed bone, but a dramatic increase in aorta, when comparing 4.5-months-old mice with 1.5-month-old ones (Fig. 2d).

However, GFP signals dropped in aorta and skeletal muscle but not in bone of diabetic mice induced by streptozotocin (STZ; Fig. 2e), suggesting that EC-OCN expression is sensitive to metabolic stress. To further confirm the induction of OCN by LRP1 depletion in ECs, we generated a LRP1 eKO; hOC-GFPtpz mouse model (Fig. 2f). As expected, GFP level was increased in aorta and skeletal muscle, but not in bone, of LRP1 eKO; hOC-GFPtpz mice, compared to their littermate control (WT; hOC-GFPtpz) mice (Fig. 2f). Collectively, our data suggest that EC is a key source of OCN in circulation during adulthood and its expression is regulated by LRP1 and metabolic stress.

**LRP1 depletion induces OCN through promoting FoxO nuclear export in ECs.** To delineate the regulatory mechanisms by which endothelial LRP1 depletion promotes OCN expression, we performed GSEA motif analysis with the mRNA-seq datasets. Our motif analysis results demonstrate that many transcription factors were potentially regulated by LRP1 depletion (Supplementary Fig. 3a). Particularly, many genes identified from

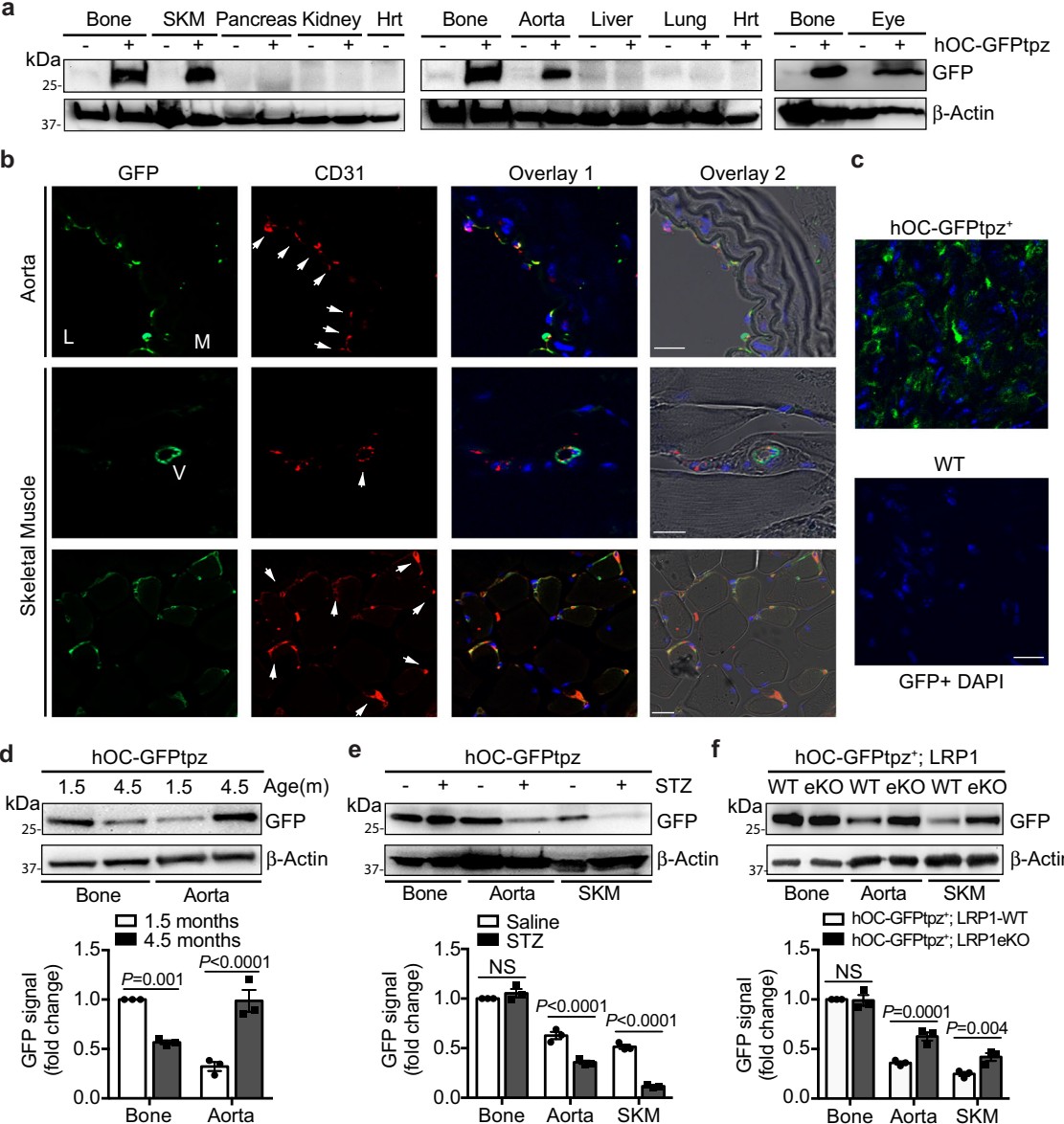

**Fig. 2 The ocn promoter-driven GFP expression in ECs. a–c** The *ocn* promoter-driven GFP expression was detected in aorta and other tissues of hOC-GFPtpz mice, determine by Western blotting (**a**), cross-section staining with aorta and skeletal muscle (**b**) and *en face* staining with aorta (**c**). SKM, skeletal muscle. Hrt, heart. Arrows indicate GFP-positive ECs. L, lumen. M, media. V, vessel. Negative control images with tissues of non-transgenic mice are shown in Supplementary Figure 2j. **d** GFP levels in bone and aorta isolated from hOC-GFPtpz mice at indicated age. **e** GFP levels in bone, aorta and skeletal muscle isolated from streptozotocin (STZ)-induced diabetic or control hOC-GFPtpz mice. **f** GFP levels in bone, aorta and skeletal muscle isolated from LRP1 eKO; hOC-GFPtpz or WT; hOC-GFPtpz mice. Scale bar, 20 μm. $n = 3$ (**d–f**). NS, not significant. Data are presented as mean ± SEM. Analysis was two-way ANOVA followed by Fisher's LSD multiple comparison test (**d–f**).

mRNA-seq assays contained upstream consensus binding sites for FoxOs (Supplementary Fig. 3b), a subgroup of the Forkhead family of transcription factors[22]. In osteoblasts, FoxO1 is responsible for suppressing OCN expression and inducing OST-PTP expression, which further blocks OCN bioactivity[23,24]. Therefore, we hypothesize that LRP1 regulates OCN expression through regulating FoxO repressor activity in ECs. LRP1 is a heterodimer containing an extracellular α chain and a membrane-anchored cytoplasmic β chain (LRP1β). Upon certain stimuli, LRP1β can also be processed by γ-secretase and translocated to the nucleus, where it regulates the activity and subcellular localization of nuclear enzymes and transcriptional regulators[11,14,25,26]. Given that LRP1β regulates the activity of these nuclear proteins mainly through protein-protein

interaction, we started to test whether LRP1 can interact with FoxOs. Indeed, both overexpressed and endogenous LRP1β were associated with FoxO1, FoxO3 and FoxO4, demonstrated by immunoprecipitation assays (Fig. 3a, b). Next, we evaluated the subcellular location of LRP1 and FoxO1 in MLECs with immunofluorescence imaging and subcellular fractionation assays. In control MLECs, FoxO1 signals were observed in both the cytosol and nucleus (Fig. 3c–e). However, when LRP1 was depleted, FoxO1 signals were increased in cytosol but decreased in nucleus, indicating its nuclear export (Fig. 3c–e). Next, we tested whether overexpression of constitutively active FoxO1 (CA-FoxO1), where AKT phosphorylation sites were mutated[27], could 'rescue' the inhibitory effect of LRP1 on OCN induction. We performed imaging studies and observed that CA-FoxO1 signals were

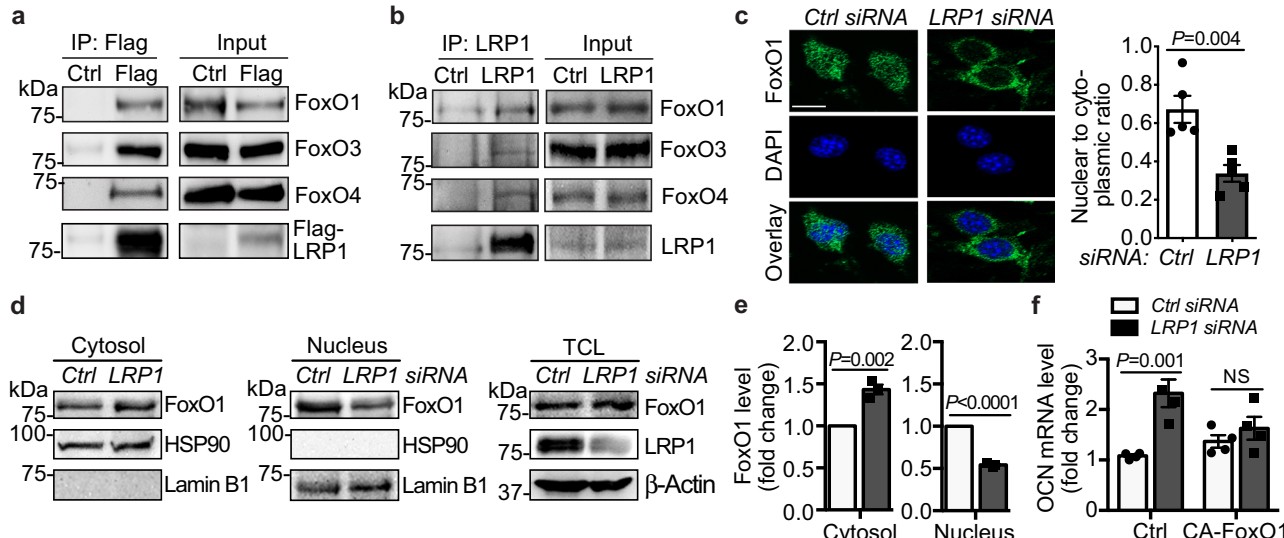

**Fig. 3 LRP1 depletion induces OCN through increasing FoxO nuclear export in ECs. a, b** LRP1 was associated with FoxOs. Lysates of HEK 293 cells containing stably expressed Flag-LRP1β (**a**) or MLECs (**b**) were immunoprecipitated with anti-Flag or anti-LRP1 resin and blotted for FoxOs. **c–e** LRP1 depletion in MLECs led to FoxO1 nuclear export. MLECs were transfected with LRP1 or control siRNAs and subjected for immunofluorescence imaging (**b**) or subcellular fractionation assays (**d, e**). MLECs were stained for FoxO1 (green) and the nucleus (DAPI, blue, **b**) and the intensity ratio of FoxO1 signals in the nucleus compared to that in cytosol was quantified. Scale bar, 10 μm. TCL, total cell lysates. **f** Constitutively active FoxO1 (CA-FoxO1) inhibited LRP1 depletion-induced OCN, analyzed with real-time PCR. n = 5 (**c**), 3 (**e**), and 4 (**f**). NS, not significant. Data are presented as mean ± SEM. Analysis was two-way ANOVA followed by Fisher's LSD multiple comparison test (**f**) or unpaired two-tailed Student's t-test (**c, e**).

localized in the MLEC nucleus even when LRP1 was depleted (Supplementary Fig. 3c). By performing real-time PCRs, we detected that LRP1 depletion-induced OCN mRNA level or OCN secreted into the conditioned media was blocked by the over-expression of CA-FoxO1 (Fig. 3f, Supplementary Fig. 3d). Taken together, these results suggest that LRP1 depletion induces OCN expression through promoting nuclear export of FoxO1.

**Endothelial LRP1 depletion improves glucose homeostasis in diabetic mice.** Similar as our previous observations with LRP1$_{Tie2}^{-/-}$ mice[14], LRP1 eKO mice displayed increased insulin sensitivity, without dramatic changes in glucose tolerance, compared to their littermate control (WT, LRP1$^{f/f}$; Cdh5-CreER$^{-/-}$) mice (Supplementary Fig. 4a, b). To further elucidate insulin action in vivo, we performed hyperinsulinemic-euglycemic clamp assays and discovered that EC-LRP1 depletion led to a significant increase in glucose uptake in skeletal muscle and a milder increase in white adipose tissue, without changes in hepatic glucose production at basal condition or after clamp (Supplementary Fig. 4c–e). These data suggest that LRP1 depletion in ECs majorly increases glucose utilization in skeletal muscle in physiological condition. In a diet-induced obese (DIO) mouse model, LRP1 eKO mice displayed less weight gain and lower blood insulin, glucose, TG and FFAs levels and HOMA-IR scores compared to WT mice following high-fat diet (HFD) feeding (Supplementary Fig. 5a–f). Improved responses in glucose clearance and insulin sensitivity were also observed in HFD-fed LRP1 eKO mice (Fig. 4a). By performing hyperinsulinemic-euglycemic clamp studies, we observed significantly higher glucose infusion rate (GIR), the amount of exogenous glucose required to maintain euglycemia, and the glucose disposal rate (GDR) in HFD-fed LRP1 eKO mice compared to WT mice (Fig. 4b, c). In addition, glucose uptake in skeletal muscle and WAT was substantially increased in LRP1 eKO mice although there was no change with hepatic glucose production (Fig. 4d, e and Supplementary Fig. 5g). It suggests EC-LRP1 depletion improves insulin sensitivity by increasing glucose uptake in peripheral tissues in DIO

mice. Next, we investigated the effect of EC-LRP1 depletion in T1DM. Following STZ-induced islet beta-cell injury, serum insulin level decreased to minimum in both LRP1 eKO and WT mice (Fig. 4f), which is consistent to markedly decreased beta-cell masses in both LRP1 eKO and WT mice compared to their non-STZ controls (Supplementary Fig. 5h). In addition, insulin level was lower in LRP1 eKO mice than WT mice before STZ injection but higher after STZ injection (Fig. 4f). However, no significant beta-cell mass differences were detected between LRP1 eKO and WT mice before or after STZ treatments (Supplementary Fig. 5h). It suggests the regulation of insulin level in LRP1 eKO mice involves some unknown mechanisms that still need further investigation. Interestingly, LRP1 eKO mice were less hyperglycemic than WT mice, although their body weights were similar (Fig. 4g, h). Clamp studies demonstrated that LRP1 depletion decreased hepatic glucose production at the basal and clamp conditions and increased glucose infusion rate and glucose uptake in skeletal muscle (Fig. 4i–m), suggesting an improved glucose response in LRP1 eKO mice upon insulin deficiency. Taken together, our results suggest that EC-LRP1 depletion improved glucose homeostasis in diabetic mice.

**OCN requires IGF1R and GPRC6A to activate insulin signaling pathway.** OCN is an osteoblast-secreted hormone to stimulate pancreatic beta-cell proliferation and insulin secretion, and insulin sensitivity in liver, muscle and white adipose tissue[16–18,28]. However, the underlying molecular mechanisms by which OCN improves insulin sensitivity are not fully understood. Insulin signaling pathway is a central player in tuning blood glucose levels[29]. Therefore, we investigated whether OCN could activate the insulin signaling pathway in metabolic tissues. Similar as insulin and IGF1, OCN increased phosphorylation of IRS1, AKT and GSK3β in mouse skeletal muscle and liver, and induced membrane translocation of GLUT4 in skeletal muscle (Fig. 5a, b and Supplementary Fig. 6a–c), indicating the activation of insulin signaling and glucose transport. Previous studies demonstrate that OCN induces AKT phosphorylation and GLUT4 membrane

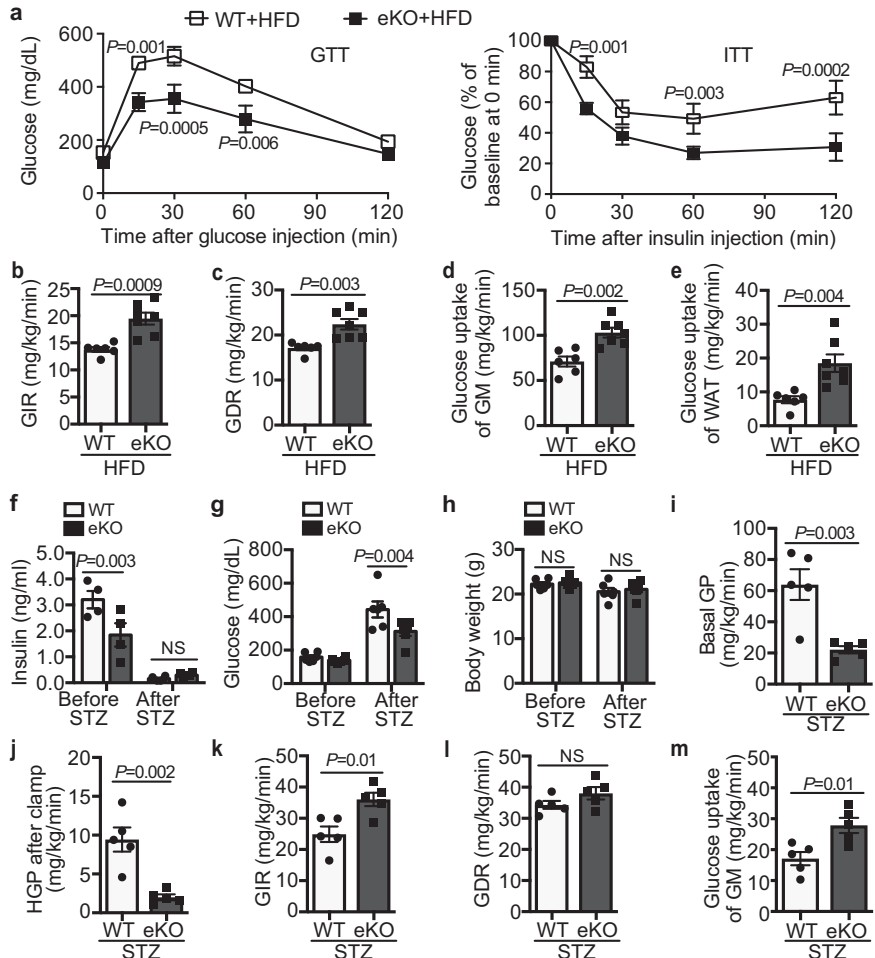

**Fig. 4 LRP1 eKO mice display improved glucose response in diabetic mice.** Glucose studies were performed with HFD-fed (**a**–**e**) or STZ-injected (**f**–**m**) LRP1 eKO and WT mice. **a** Glucose and insulin tolerance tests (GTTs, ITTs). **b**–**e** Hyperinsulinemic-euglycemic glucose clamp studies were performed in HFD-fed WT and LRP1 eKO mice for measurements of (**b**) GIR, (**c**) GDR, glucose uptake in gastrocnemius muscle (GM, **d**) and white adipose tissue (WAT, **e**). Results for control chow-fed mice are included in Supplementary Fig. 4. **f**–**h** Blood levels of insulin (**f**), glucose (**g**) and body weight (**h**). **i**–**m** Hyperinsulinemic-euglycemic glucose clamp studies were performed in WT and LRP1 eKO mice after STZ-induced diabetes for measurements of basal glucose production (GP, **i**), hepatic GP after clamp (HGP, **j**), GIR (**k**), GDR (**l**), and glucose uptake of GM (**m**). $n = 7$ (**a**; **b**–**e**, eKO), 6 (**b**–**e**, WT; **f**, WT, after STZ; **g**–**h**), 4 (**f**, WT, before STZ; **f**, eKO), 5 (**i**–**m**). NS, not significant. Data are presented as mean ± SEM. Analysis was two-way ANOVA followed by Fisher's LSD multiple comparison test (**a**, **f**–**h**) or unpaired two-tailed Student's $t$-test (**b**–**e**, **i**–**m**).

translocation in muscles[30] and our data confirmed these findings. To understand how IRS1 is phosphorylated by OCN, we evaluated the autophosphorylation of IR and IGF1R, an indicator for the activation of IR and IGF1R. Interestingly, OCN increased phosphorylation of IR and IGF1R at tyrosine 1131 and 1146, respectively (Fig. 5a, b and Supplementary Fig. 6a, b), suggesting that OCN promotes the activation of IR and IGF1R. OCN can signal through GPRC6A[30,31]. Therefore, we tested whether OCN and GPRC6A could form a complex with IR or IGF1R and whether they are required for the activation of insulin signaling downstream mediators. By performing co-immunoprecipitation assays, we observed that Flag-tagged OCN was co-immunoprecipitated with overexpressed IGF1R or IR, and GPRC6A was also in the complex with IGF1R in HEK293 cells (Fig. 5c–f). In addition, endogenous IR and IGF1R were also detected in the complex with OCN in primary hepatocytes (Fig. 5g). In IGF1R$^{-/-}$ hepatocytes, OCN-induced phosphorylation of IRS1, AKT and GSK3β was inhibited dramatically (Fig. 5h and Supplementary Fig. 6d), suggesting that IGF1R is required for OCN-induced insulin signaling. In addition, IGF1R depletion by its siRNAs significantly decreased the amount of 2-deoxyglucose

(2DG) uptake in C2C12 myoblasts (Fig. 5i and Supplementary Fig. 6e). GPRC6A depletion with its specific siRNAs in hepatocytes also blocked IRS1 phosphorylation induced by OCN, but not by insulin or IGF1 (Fig. 5j and Supplementary Fig. 6f), suggesting that GPRC6A is required for OCN-induced IRS1 activation. Taken together, these results indicate that OCN promotes the activation of the insulin signaling likely through GPRC6A and IGF1R.

**OCN, induced by LRP1 depletion in HFD-fed mice, promotes blood glucose clearance.** OCN knockout mice are obese, displaying glucose intolerance and insulin resistance[17], which is opposite to the metabolic phenotype of LRP1 eKO mice where blood OCN level was increased (Fig. 1d). In addition, blood OCN levels are inversely correlated with the metabolic syndrome, adiposity and insulin resistance[32]. Similarly, we also observed decreases of blood OCN levels in metabolic syndrome patients (Fig. 6a). To further determine how LRP1 depletion-induced OCN changes in diabetic conditions, we measured blood OCN levels in HFD-fed LRP1 eKO and WT mice. As expected, blood Glu-, Gla- and total OCN levels were decreased in HFD-fed WT

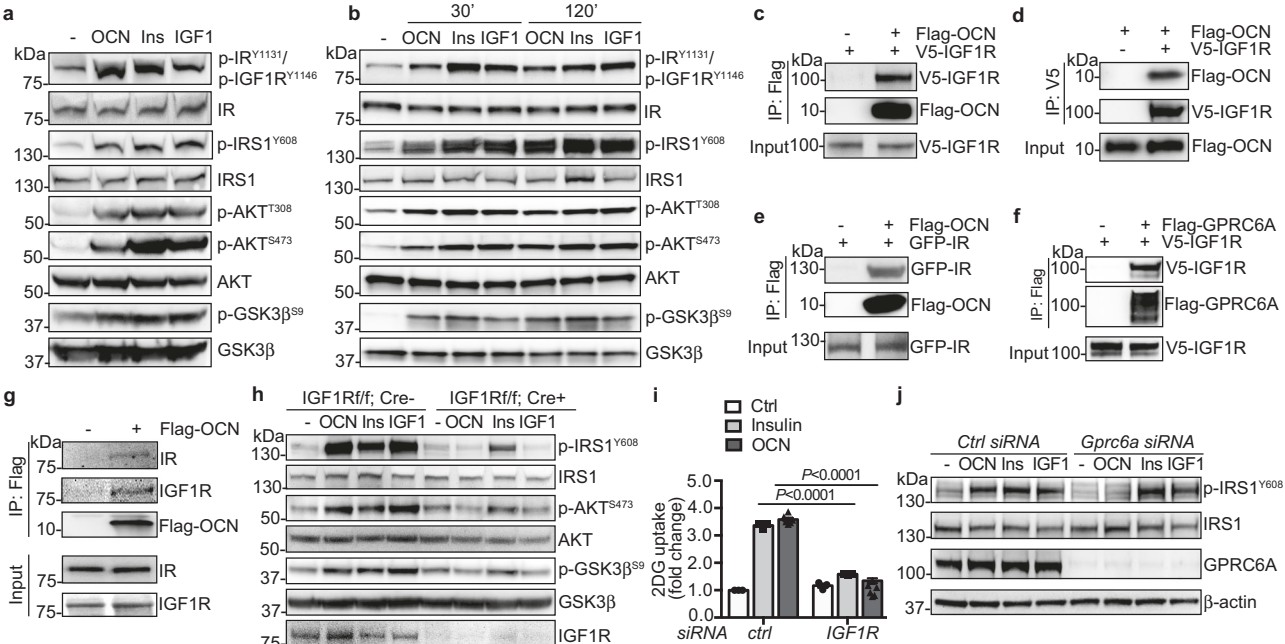

**Fig. 5 OCN requires GPRC6A and IGF1R for the activation of the downstream insulin signaling pathway. a, b** OCN promoted insulin signaling in skeletal muscle (**a**) and liver (**b**). Ins, insulin. **c–f** OCN was associated with IGF1R (**c, d**) and IR (**e**), and GPRC6A was associated with IGF1R (**f**) in HEK 293 cells. **g** OCN was in the complex with endogenous IR and IGF1R in primary hepatocytes. **h** IGF1R was required for OCN-induced phosphorylation of insulin signaling mediators in primary hepatocytes. **i** IGF1R knockdown inhibited OCN-promoted 2DG uptake. C2C12 cells were transfected with IGF1R siRNAs and then treated with OCN or insulin for 2 h. $n = 3$ for three independent repeats of each experiment. **j** GPRC6A knockdown inhibited OCN-induced phosphorylation of insulin signaling mediators in primary hepatocytes. $n = 3$ (**i**, ctrl, Insulin), 6 (**i**, OCN). Data are presented as mean ± SEM. Analysis was two-way ANOVA followed by Fisher's LSD multiple comparison test (**i**).

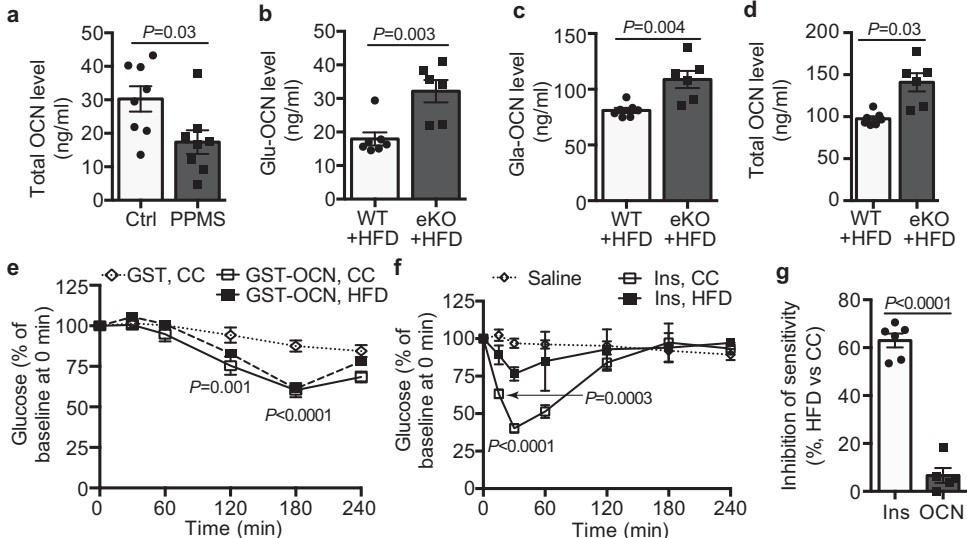

**Fig. 6 OCN, induced by LRP1 depletion in HFD-fed mice, promotes blood glucose clearance. a** Blood OCN levels in human metabolic syndrome patients (MS) and normal lean controls (Ctrl). **b–d** Blood Glu-, Gla- and total OCN levels in HFD-fed WT and eKO mice. Results for control chow (CC)-fed mice are included in Fig. 1d. **e–g** OCN and insulin tolerance tests in CC and HFD mice (OTTs in **e**, ITT in **f**). The inhibition of maximal sensitivity for insulin and OCN in HFD-fed mice compared to CC-fed ones is presented in **g**. $n = 11$ (**a**), 7 (**b–d**, WT), 6 (**b–d**, eKO), 5 (**e**; **f**, saline; **g**, OCN), 4 (**f**, insulin, CC), 6 (**f**, insulin, HFD; **g**, insulin). Data are presented as mean ± SEM. Analysis was unpaired two-tailed Student's *t*-test (**a–d, g**) or two-way ANOVA followed by Fisher's LSD multiple comparison test (**e**, **f**).

mice (Fig. 6b–d) compared to CC-fed mice (Fig. 1d). However, LRP1 depletion-induced OCN levels sustained at a high level in HFD-fed LRP1 eKO mice (Fig. 6b–d). Our signaling data (Fig. 5) suggested that OCN promotes glucose disposal by increasing the activity of insulin signaling pathway. Therefore, we evaluated whether acute administration of OCN could sufficiently promote

glucose clearance in vivo. To test this, we performed OCN tolerance tests (OTTs) during which the acute changes of blood glucose level were monitored in response to OCN injection. Interestingly, OCN injection resulted in a glucose-lowering response with the peak at 3 h after injection, which was slower than insulin (peaking at 0.5 h, Fig. 6e, f). In HFD-induced insulin-

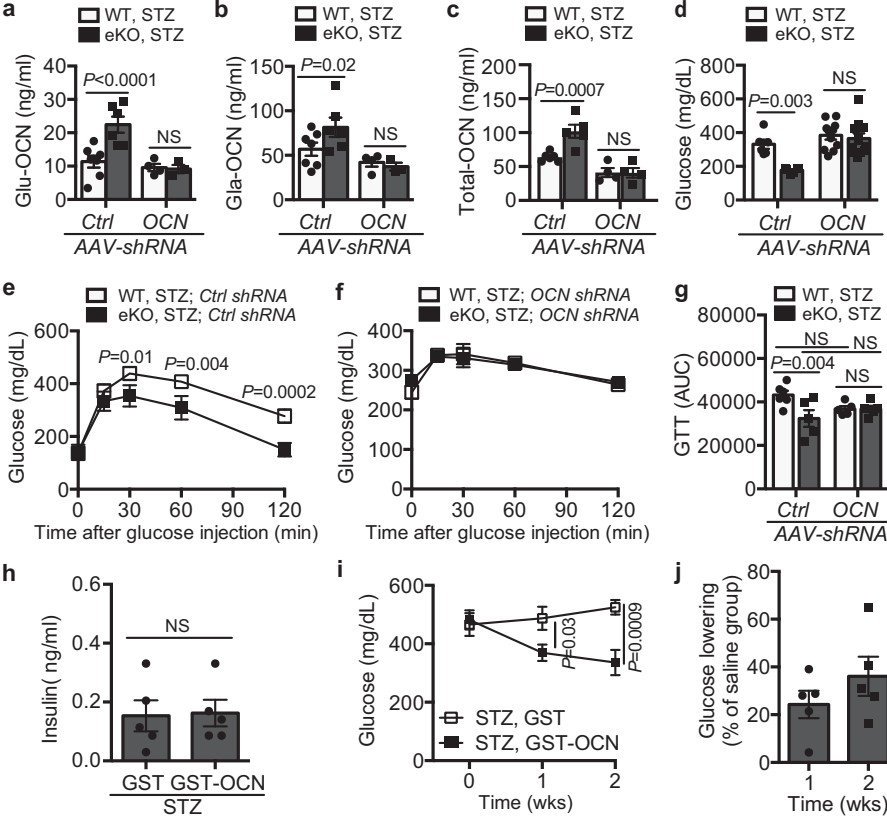

**Fig. 7 OCN silencing in vivo inhibits endothelial LRP1 depletion-improved glucose responses. a–d** Blood levels of Glu- (**a**), Gla- (**b**) and total OCN (**c**), and non-fasting glucose (**d**) after AAV-OCN-shRNA or AAV-Ctrl-shRNA injection in mice after STZ treatment. The data for control mice are shown in Fig. 1d. **e–g** Glucose tolerance tests in AAV-Ctrl-shRNA (**e**) and AAV-OCN-shRNA (**f**) injected mice after STZ-induced diabetes. **g** Areas under the curve (AUCs) for GTTs. **h** Blood levels of insulin in saline (containing GST-OCN or GST)-injected STZ mice. The data for non-STZ control mice are shown in Supplementary Fig. 5b. **i–j** OCN administration alleviates hyperglycemic response in STZ-induced type 1 diabetic mice. The percentage of glucose-lowering in GST-OCN-injected STZ mice compared to GST-injected mice is presented in **j**. n = 6 (**a–c**, WT; **d**, Ctrl-shRNA; **e–g**, WT; **h**), 5(**a–c**, eKO, Ctrl-shRNA; **e–g**, eKO; **i, j**), 4 (**a–c**, eKO, OCN-shRNA), 11 (**d**, OCN-shRNA). NS, not significant. Data are presented as mean ± SEM. Analysis was two-way ANOVA followed by Fisher's LSD multiple comparison test (**a–g, i**) or unpaired two-tailed Student's t-test (**h, j**).

resistant mice, insulin at the same dose used for CC-fed mice failed to decrease blood glucose levels in HFD-fed mice, showing a 60% decrease in its maximal sensitivity (Fig. 6f, g). However, OCN still displayed an efficient glucose-lowering response in HFD-fed mice compared to CC-fed mice (Fig. 6e, g).

**EC-LRP1 depletion normalizes hyperglycemia through OCN.**
Next, we also monitored blood OCN level in in STZ-induced diabetes and detected decreases of Glu-, Gla- and total OCN levels in diabetic mice (Fig. 7a–c compared to Fig. 1d). However, EC-LRP1 depletion normalized these decreases (Fig. 7a–c). To elucidate whether OCN is required for EC-LRP1 depletion-improved glucose responses in diabetic mice, we depleted OCN protein with adeno-associated viral particles containing OCN shRNA (OCN AAV-shRNA) in vivo. As expected, OCN AAV-shRNA decreased blood OCN levels in LRP1 eKO mice to a similar level as that in WT mice (Fig. 7a–c), indicating the successful knockdown of OCN in mice. We compared blood glucose levels between OCN AAV-shRNA or control virus-injected mice. The results showed OCN AAV-shRNA-injected LRP1 eKO mice displayed severe hyperglycemia, displaying a reverse response to the normalized non-fasting glucose level in LRP1 eKO mice following STZ injection (Fig. 7d). During glucose tolerance tests, STZ-injected LRP1 eKO mice also displayed more efficient glucose clearance than WT mice (Fig. 7e, g). However, this improvement was abolished in OCN AAV-shRNA-injected LRP1

eKO mice (Fig. 7f, g). Taken together, our results suggest OCN is required for EC-LRP1 depletion to protect mice from T1DM. Last, we tested whether OCN administration is sufficient to alleviate hyperglycemia in mice with T1DM when insulin is deficient. Following daily injection of OCN for two weeks, no changes were observed with blood insulin levels (Fig. 7h). However, blood glucose levels markedly dropped within two weeks, with a 29.3±7.3% or 41.0±16.9% decrease at one or weeks after OCN injection, compared to saline group (Fig. 7i-j). Taken together, these data suggest that OCN efficiently improves glucose response in both insulin resistant and deficient mice and may possess a great potential to treat diabetes.

**Discussion**
In this study, we show that LRP1 in the vascular endothelium plays a critical role in defining whole-body glucose metabolism through secreting OCN. In the absence of endothelial LRP1, the secretion of OCN from ECs is increased, which improves glucose uptake of skeletal muscle and WAT and decreases glucose production of hepatocytes. When OCN is depleted, the protective effects of endothelial LRP1 depletion in diabetes are abrogated. Based on these results, we propose a signaling cascade induced by endothelial LRP1 depletion for the regulation of glucose metabolism. Specifically, LRP1 depletion in ECs leads to FoxO1 nuclear export, which relieves the inhibitory effect of FoxO1 on OCN induction. Once OCN is secreted into circulation, it

promotes the activation of insulin signaling cascades and glucose uptake. This signaling pathway establishes a direct cause-effect relationship between vascular endothelium and glucose homeostasis through endocrine/paracrine regulation. In addition, our findings indicate that OCN might act as an endogenous stimulus for insulin signaling and provide insights into treating diabetes and insulin resistance.

Our understanding of ECs, which were initially recognized as a thin layer of squamous cells lining in the inner surface of the circulatory system, has evolved significantly in the last two decades. It has been well appreciated that ECs play active roles in a variety of biological processes. However, most research has focused on the regulatory roles of endothelial dysfunction in vascular diseases in response to injury or stress such as diabetes[33]. Based on the emerging data that we present here, we start to appreciate that the crosstalk between ECs and metabolic tissues/cells is bidirectional[34]. A group of EC-derived regulatory factors have been reported, such as nitric oxide, insulinotropic factors and growth factors that increase insulin secretion[35,36]. The discovery that OCN can be secreted by ECs provides evidence for the endocrine/paracrine regulation of glucose homeostasis by the vascular endothelium. Therefore, we propose that endothelial dysregulation should be considered as an important contributor to abnormal glucose handling. Our previous studies demonstrate critical roles of LRP1 in endothelial cell function, such as angiogenesis, inflammation and lipid transport[11–14]. These processes regulated by LRP1 might also contribute to the improvement of glucose metabolism in LRP1 eKO mice, which warrants further investigation. Before the discovery of OCN induction in ECs, OCN was considered to be secreted mainly from osteoblasts[16–18,28]. Recent studies indicate that OCN expression can be detected in bone marrow-generated procalcific cells such as monocytes, endothelial progenitor cells and megakaryocytes[37–40]. Our data suggest that vascular endothelium could be another important source for OCN. In our studies with AAV-OCN shRNA injection, OCN depletion was not limited to ECs. Therefore, the "rescue" effect for AAV-OCN shRNA could be maximized due to OCN knockdown in a variety of cells (i.e., ECs, osteoblasts). In osteoblasts, the FoxO-Runx2 signaling pathway is responsible for insulin-induced OCN expression[24]. Likewise, we also show that LRP1 depletion induces OCN through promoting FoxO nuclear export. It indicates that similar signaling cascades induce OCN in ECs and osteoblasts. Given that OCN-driven GFP expression is decreased in aorta and skeletal muscle but not in marrow-flushed bone (Fig. 2e), we speculate that EC-specific regulatory mechanisms for OCN induction also exist and remains to be further characterized. The hOC-GFPtpz reporter mouse model has been validated as a great tool for the understanding of human OCN promoter activation[15,41]. Based on our observations in hOC-GFP transgenic mice (Fig. 2e, f), the regulation of human Ocn promoter-driven GFP expression by hyperglycemia and LRP1 depletion is similar as that of mouse OCN. Interestingly, OCN mRNA and protein were detected in mouse lung, heart and liver ECs (Fig. 1) but human Ocn promoter-driven GFP signals were not observed in these organs (Figs. 1 and 2). It suggests the expression pattern of human OCN might be different from mouse OCN. In addition, its expression in mouse tissues other than lung, heart and liver still need further evaluation due to EC heterogeneity. The mouse Ocn promoter-driven reporter mouse models would be a great tool for these studies.

Previous studies demonstrate that OCN increases insulin sensitivity through indirect mechanisms, including increasing secretion of adiponectin or reducing ER stress through NFκB and PI3K signaling[17,42]. In addition, OCN increases glucose and fatty acid utilization in myofibers through upregulating interleukin 6, resulting an enhancement of exercise capacity[30]. Long-term (28 days) administration of OCN enhances insulin-stimulated downstream signaling in liver, adipose tissue and skeletal muscle[42]. We observed that OCN also acutely activated IRS1 in vivo (Fig. 5a, b). It suggests that the protective effect of OCN in glucose homeostasis involves in multiple layers of regulation. A putative receptor for OCN is GPRC6A[30,31]. Both GPRC6A[-/-] and OCN[-/-] mice show similar phenotypes, including exacerbated glucose intolerance and impaired insulin secretion[17,43,44]. OCN can activate GPRC6A in a dose-dependent manner[31]. The binding of OCN to the cell is blocked by GPRC6A depletion[43]. We discovered that both OCN and GPRC6A were in the complex with IGF1R and the depletion of IGF1R or GPRC6A blocked OCN-induced downstream insulin signaling events (Fig. 5c–j), suggesting that OCN promotes insulin signaling pathway through GPRC6A and IGF1R. However, how OCN signaling leads to the activation of insulin signaling pathway and what are the specific roles of GPRC6A, IGF1R and IR in this signaling cascade remain to be further determined.

Insulin therapy is the standardized treatment for T1DM, however, many patients with T1DM still display A1C levels higher than 7.0% and poor metabolic control[45]. Standard insulin therapy in T1DM is associated with increased complications, including hypoglycemia, weight gain, dyslipidemia, and insulin resistance[46]. Therefore, combined therapies that are used for T2DM are also needed for T1DM at certain conditions[46]. IGF1 has been tested for treating both T1DM and T2DM. However, its side effects (i.e., cancer risk) limit its potential application as a hypoglycemic agent. Our studies suggest that OCN could improve glucose tolerance in both insulin-deficient and insulin-resistant conditions. However, adverse effects might exist for OCN since it is often associated with vascular calcification[47,48]. Therefore, understanding the underlying mechanisms of OCN regulation of glucose metabolism and dissecting its interplay with insulin and IGF1 will shed light on how to use it to treat diabetes in an effective and safe manner.

## Methods

**Mice and subjects.** We used the mating of LRP1[flox/flox] and Cdh5-CreER[+/-] mice to generate the LRP1[f/f]; Cdh5-CreER[±/-] (WT or eKO) male mice for normal chow, HFD feeding and STZ-induced diabetes studies. hOC-GFPtpz transgenic mice were crossed with LRP1 eKO mice to examine the *ocn* promoter-driven GFP expression. B6; 129S7-Lrp1[tm2Her]/J (LRP1-floxed; LRP1[f/f]), B6.Cg-Tg (CAG-Cre/Esr1*) 5Amc/J (CAG-CreER[+/-]) and B6;129-Igf1r[tm2Arge]/J (IGFIR-floxed; IGF1R[f/f]) mice were obtained from Jackson Laboratories. The Cdh5 (PAC)-CreERT2 (Cdh5-CreER[+/-]) mice were generously provided by Dr. Ralf H. Adams[49]. All adult mice were fed the control chow (CC, 14.7% calories from fat) or HFD (60% calories from fat) for 12 weeks. For STZ-induced diabetes, adult mice were injected with STZ (40 mg/kg, intraperitoneally; *i.p.*) for 5 consecutive days. Blood serum was obtained before and after they were fed with different diets. Primary MVECs were isolated from mouse liver, lung and heart. For insulin signaling experiments in Fig. 5a, b and 5h, mice were injected with insulin at 0.5 U/kg, IGF1 at 800 μg/kg and OCN (Abcam) at 150 μg/kg for indicated time period. In addition, STZ mice were injected daily with saline containing GST-OCN or GST at 100 μg/kg for 2 weeks and blood glucose levels were monitored. Adenoviral particles expressing mouse OCN shRNA and control shRNA-contained virus were injected into the mice via tail vein with a titer at $7 \times 10^{11}$ per 25 g mice. All mice were housed on a 12-h light/dark cycle, with food and water ad libitum. Mouse rooms were maintained at 65–75 °F (~18–23 °C) with 40–60% humidity. All experimental procedures on mice were performed according to the National Institutes of Health Guide for the Care and Use of Laboratory Animals and approved by the Institutional Committee for the Use of Animals in Research at Baylor College of Medicine. Studies also performed with human samples collected through the study[50] that has been approved by the Institutional Review Board for Baylor College of Medicine and Affiliated Hospital. Informed consent was obtained from all subjects. The detailed baseline characteristics (i.e., age, BMI, sex, blood glucose) of these participants have been listed in the Table 1 of the previously published paper[50].

**Cell lines and primary cells, immunoblotting, immunoprecipitation, and membrane fractionation.** HEK293 cells were grown in Dulbecco's modified Eagle medium (DMEM) supplemented with 10% fetal bovine serum (FBS) and

antibiotics (100 U/ml penicillin, 68.6 mol/l streptomycin). Mouse primary microvascular ECs were isolated using PECAM-1 antibody Dynabead selection[14]. C2C12 cells were cultured in DMEM supplemented with 10% FBS, 5% horse serum and antibiotics (100 U/ml penicillin, 68.6 mol/l streptomycin). Primary hepatocytes were isolated using collagenase perfusion method[51]. For immunoprecipitation (IP) experiments, protein A/G Plus-agarose was used to pull down antibody complexes following our established methods[11]. For transient transfection, HEK293 cells were transfected with V5-tagged IGF1R, GFP-IR or Flag-GPRC6A plasmids with Lipofectamine 2000. Two days later, HEK293 cells or primary hepatocytes were treated with Flag-tagged OCN protein and then washed with cold PBS and cross-linked with DSP (dithiobis(succinimidyl propionate)) at 4 °C for 2 h. Cell lysates were harvested and IPed with anti-Flag resin and precipitates were blotted with indicated antibodies. Membrane fractionation was performed based on our published protocol with small modification[13].

**Analysis of endocrine hormones and metabolites.** Serum values for glucose were measured with a mouse endocrine multiplex assay, and insulin with ELISA kits. Free fatty acid (FFA) assays were performed with non-esterified fatty acids kits. Lipid contents were measured with Infinity triglyceride kits.

**Glucose/insulin/ OCN tolerance tests (GTTs, ITTs, OTTs).** Glucose tolerance tests (GTTs) were performed after an overnight (for CC and HFD-fed mice) or 6 h fasting (for STZ-induced diabetes mice) following the published protocol[52]. Blood glucose was measured after an i.p. glucose injection (1 g/kg) with a Freestyle Glucose Monitoring System (Abbott Laboratories). Insulin (ITTs) and OCN tolerance testing (OTTs) were performed after 4 h fasting. Blood glucose was measured after an intravenous injection of OCN (150 µg/kg, Abcam) for OTTs, and an i.p. insulin injection at 0.5 U/kg for ITTs in CC-fed (Supplementary Fig. 4a) and HFD-fed mice (Fig. 6f), and 1 U/kg for HFD-fed mice (Fig. 4a).

**Hyperinsulinemic-euglycemic clamp.** The studies were performed in unrestrained mice using the insulin clamp technique (using variable insulin doses) in combination with [3H]glucose and [14C]2-deoxyglucose following the published protocol[52]. In summary, mice were cannulated and allowed to recover for 4 to 7 days before the clamp. After an overnight fasting, mice received a primed dose of [3H]glucose (10 µCi) and then a constant rate intravenous infusion (0.1 µCi/min) of [3H]glucose using a syringe infusion pump for 90 min. Blood samples were collected for the determination of basal glucose infusion. After 90 min, mice were infused with insulin for 2 h (4 or 10 milliunits/kg/min for control chow-fed or high-fat diet-fed mice, respectively). Simultaneously, 25% glucose was infused at an adjusted rate to maintain the blood glucose level at 100–140 mg/dL. Blood glucose concentration was determined every 10 min by a glucometer. At the end of a 120-min period, blood was collected for the measurements of hepatic glucose production and peripheral glucose disposal rates. For tissue specific uptake, we inject 2-deoxy-D-[1,-14C] glucose (10 µCi) into bolus during hyperinsulinemic-euglycemic clamp at 45 min before the end of the clamps and collect blood sample at 5, 10, 15, 25, 35, and 45 min. At the end of the clamp, mouse tissues were harvested for the evaluation of glucose uptake.

**Gene expression analysis (real-time PCR and RNA-seq).** Total RNAs were reversely transcribed into cDNAs with iScript™ cDNA synthesis kit. The real-time PCR was performed with FastStart Universal Probe Master mix, specific primers (Supplementary Table 2) and probes for each gene (Universal ProbeLibrary Probes #97 for LRP1, #32 for OCN, #71 for OCN1 and OCN2, #13 for Selplg, #7 for Alox5ap, #80 for HMGCS2, #5 for Timp4 and #80 for GAPDH) in Roche Light-cycler 480 PCR machine. Reaction mixtures were incubated at 95 °C for 10 min followed by 55 cycles at 95 °C for 10 s and 60 °C for 30 s. GAPDH was used as the housekeeping gene. Total RNAs isolated from mouse lung, heart and liver ECs were processed to generate cDNA libraries that were sequenced on the Illumina HiSeq 2500 platform by the Baylor RNA Profiling Core laboratories. Sequencing data were trimmed using trimGalore and mapped to mouse genome build UCSC mm10 using the STAR software version 2.7.1. Gene expression was quantified using featureCounts version 1.6.4 using the GENCODE gene model. Differential expression was obtained using the R package EdgeR version 3.32.0, with significance achieved at a fold-change of 2.0 and FDR ( <0.05)-adjusted p-value <0.05. Gene set enrichment analysis was performed using the GSEA software against the Molecular Signature database. The pathway collections KEGG, Reactome, Hallmark, and Gene Ontology Biological processes were used to determine enriched pathways.

**Designed siRNAs and transient transfection.** The stealth siRNA duplexes against human LRP1 were obtained from Life Technologies. The siRNA is a duplex of 5′- GGGUGGAGAGUAACCUGGAUCAGAU-3′. The control siRNA is the Stealth RNAi negative control duplex (Cat. No. 12935-300) and was purchased from Life Technologies. The siRNA duplexes against mouse IGF1R and GPRC6A were obtained from Santa Cruz Biotechnology. The siRNAs were transfected into isolated wild-type ECs according to our previous published protocol[25]. Briefly, for each sample, $2 \times 10^5$ ECs were transfected with 100 pmol siRNA. Experiments with siRNA-transfected ECs were performed 2 days later. CA-FoxO1 (pcDNA3 Flag

FKHR AAA mutant containing mutations of T24A, S256A and S319A) was transfected in ECs following our previous published protocol[11].

**Immunofluorescent studies.** Immunostaining with frozen sections, fresh tissues and cultured cells was performed following the previous protocols[14,53]. Aorta, skeletal muscle, and eyes were fixed overnight in 4% PFA in PBS. Following the incubation with 15 and 30% sucrose gradient, tissues were frozen in OCT mounting media. Cryosections of 5 µm thickness were processed for staining with indicated antibodies. For en-face staining of aorta, aortic segments were dissected out and gently cleaned of the adventitia and fixed in 3.7% formaldehyde for 10 min at room temperature, followed by staining with indicated antibodies following the previous protocol[53]. The en face images of the endothelial layer and the cross-sectional images of tissues were visualized by confocal laser scanning microscopy. To quantify the co-localization of LRP1 and CA-FoxO1, each stack optical section was analyzed using the Coloc 2 plug-in (Fiji, https://imagej.net/Coloc_2) developed for ImageJ software following the previous protocol[14]. The co-localization of LRP1 with CA-FoxO1 was estimated by the use of Pearson's correlation coefficient.

**Glucose uptake assay.** C2C12 cells were cultured and used to determine glucose uptake using a Glucose Uptake Assay Kit (colorimetric) and Glucose Uptake-Glo™ Assay Kit according to the manufacturer's protocols.

**Purification of OCN recombinant protein.** The OCN plasmid was transformed into E.coli Lemo21 (DE3) cells. The culture was grown overnight with vigorous shaking and then added 1000 µM L-rhamnose. Cells were then cultured at 30 °C until O.D. reached 0.6. IPTG (400 µM) was added and the culture was incubated at 30 °C overnight. The bacteria culture was harvested and resuspended in 20 mM Tris pH 7.5, 0.1 % Triton X-100 buffer containing protein inhibitor cocktail. The mixture was sonicated and then centrifuged at 200,000 x g (Ti45 rotor) at 4 °C for 30 min. The clarified lysate was mixed with glutathione-Sepharose 4B (AP Biotech) at 4 °C for 2 h. Beads were washed several times with 20 mM Tris pH 7.5, 0.1 % Triton X-100, followed by a single wash with detergent-free 50 mM Tris pH 8.0 buffer. GST-fusion proteins were eluted with 10 mM reduced glutathione in 50 mM Tris pH 8.0 solution. The eluted protein was dialyzed overnight against PBS. The PBS buffer was changed several times during the dialysis. Then the elution was loaded onto SDS-PAGE gel and the purchased OCN protein (Abcam) was loaded as a positive control. The gel was stained with Coomassie blue G250 for quantification and also detected for OCN with Western blotting. In addition, HEK293 cells were transfected with OCN with Flag-tag at its c-terminus. Then, their conditioned media were collected for Flag-OCN protein enrichment through anti-flag resin followed by elution with Flag peptide and passing through filters with cutoff size at 4 kDa. These GST-OCN proteins were used for in vivo experiments and Flag-OCN proteins were used for biochemical studies with cultured cells.

**Isolation of bone and osteoblasts from mice.** Tibia and femur marrow-flushed bone were collected from 4~6 weeks old mice and soft tissues were removed. The crushed bone pieces were either lysed in the protein lysis buffer or Trizol solution (Thermo Fisher) for protein or RNA purification. Osteoblasts were isolated from mouse calvaria based on a published protocol with minor modifications[54]. Briefly, the mouse calvaria bone was collected and the surrounding soft tissue was removed. Bone was chopped into small fragments of 1–2 mm² and incubated for 30 min in a shaking water bath at 37 °C with dissociation solution (DMEM with final concentration of 2.7 units/ml collagenase class I and 11.8 units of collagenase class II), followed by second incubation with dissociation solution for 30 min, one incubation with Trypsin (5 mg/ml in PBS, EDTA 0.1 g/ml) for 30 min and third incubation with dissociation solution for 30 min. After three washes with DMEM and 10% FBS, Bone fragments were cultured with DMEM supplemented with 10% FBS, 100 µg/ml ascorbic acid and antibiotics (100 U/ml penicillin, 68.6 mol/l streptomycin). Two weeks later, Cells were collected for protein and RNA extraction.

**Beta-cell mass estimation.** Pancreatic sections were deparaffinized and then rehydrated through the treatments with decreased ethanol gradients. Microwave antigen retrieval using 10 mM citrate buffer (pH 6.0) was used to expose the antigens. Next, the 5-µm cryosections were blocked with 5% heat-inactivated rabbit serum for 1 h and then incubated overnight with primary antibodies against insulin diluted in the blocking solution. Sections were then incubated in the dark with a secondary antibody conjugated with Alexa Fluor 488 in blocking solution and also counter-stained with DAPI. Serial sections with each 50 µm apart were examined and imaged using immunofluorescent microscopy. Beta-cell mass was calculated as the relative beta-cell area (the percentage of insulin-positive area over total pancreatic area) multiplied by pancreatic weight following the previous protocol[17]. At least four mice were analyzed per group.

**In vivo calcein labeling and calculation of bone formation rate.** Calcein labeling and BFR calculation were performed following the previous protocols[55,56]. For dynamic measurements of bone formation, filtered calcein solution in 2.0% sodium bicarbonate (pH 7.0) were injected into 6~8 weeks old mice intraperitoneally at 7

and 2 days prior to sacrifice. Femur, tibia and vertebrae were fixed in formalin followed by 70% ethanol. Sections were analyzed under the fluorescence microscope (Zeiss). Analysis of bone formation rate (BFR) was performed with ImageJ. BFR per bone surface is the volume of mineralized bone formed per unit time and per unit bone surface. It was calculated as the product of mineral apposition rate (MAR) and mineralizing surface per bone surface (MS/BS), BFR = MAR* (MS/BS). More than 300 μm of bone surface per bone was analyzed.

**Quantification and statistical analysis.** Data are shown as mean ± SEM. "*n*" represents the number of biological replicates. Student's *t*-test (for comparison between two groups) or one-way (for comparisons of three or more groups and one variant) or two-way ANOVA (for comparisons of three or more groups and two variants) followed by Fisher's LSD post hoc pairwise tests were used for statistical analysis of data that passed for the normality of distribution by Kolmogorov–Smirnov and for equal variance. If data failed normality of distribution or equal variance tests, Mann–Whitney *U*-test (for 2-group comparison) or Kruskal–Wallis test (for comparison of three or more groups) followed by Dunnett post hoc pairwise tests was used instead. Values of *P* ≤ 0.05 were considered statistically significant. No statistical methods were used to predetermine the sample size. No randomization was used as all mice used were genetically defined, inbred mice. Data analysis for metabolic phenotype was performed in a blinded fashion. All data presented in this study are representative results of at least three independent experiments.

**Reporting summary**. Further information on research design is available in the Nature Research Reporting Summary linked to this article.

## Data availability

The data that support the findings of this study are available within the article, its Supplementary Information files. The raw sequencing data generated in this study have been deposited in the NCBI GEO database under accession number-GSE117560. The detailed information of key reagents is provided in Supplementary Table 3. The source data underlying Figures and Supplementary Figures are provided as a Source Data file. Source data are provided with this paper.

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

## Acknowledgements

We thank the Baylor Metabolism and Phenotyping Core (MMPC) and Genomic and RNA Profiling Core Laboratories for their help. We would like to thank the support from National Institute of Health (NIH) grants R01s HL112890, HL061656, and DK123186 (to X.P.), HL122736 (to L.X.), UM1HG006348 and DK114356 (to P.K.S. and MMPC), ES027544 (to Z.S.), DK111436 (to Z.S.), CA215591 (to Z.S.), American Heart Association (30970064; to Z.S.) and Taishan Scholarship (to Z.S.) for funding support. KR, DP, and CC were partially supported by The Cancer Prevention Institute of Texas (CPRIT) RP170005, NIH P30 shared resource grant CA125123, and NIEHS center grants P30 ES030285 and P42 ES027725.

## Author contributions

H.M., L.X., X.P., writing the first draft; theoretical and experimental investigation; scientific discussion; revision of the manuscript. L.L., Q.F., A.A., P.K.S., experimental investigation; scientific discussion. C.C., K.R., D.P., RNA sequencing data analysis; scientific discussion; revision of the manuscript. J.C. studies with IGFRflox/flox mice; scientific discussion; revision of the manuscript. H.W., C.M.B. studies with human samples; revision of the manuscript. Z.S. scientific discussion; revision of the manuscript.

## Competing interests

The authors declare no competing interests.
