## [Peer Review File · Nature Communications]

Reviewer comments, first round of review:

Reviewer #1 (Remarks to the Author):

In the current study, the authors generate an genetic depletion of the LDL receptor-related protein 1 (LRP1) in endothelial cells and identify osteocalcin (OCN) as an upregulated gene in the knockouts. Using a unique ocn promoter-driven report mouse, they confirm expression of this molecule in the vasculature endothelium, and go on to define the molecular pathway by which LRP1 regulates this molecule. Further, they demonstrate that OCN regulates glucose metabolism thereby identifying an important role for the endothelium in regulating insulin sensitivity. Overall, this is a careful and well written study that includes a variety of approaches and genetically modified mice to support the conclusions reached. There are a few issues that the authors need to address:

1. The selection criteria for the RNA seq data presented in Figure 1a and b and in supplementary Figure 2 are not very stringent (i.e. p values <0.05, FC>1.25). The authors should correct the p values for false discovery rates and use FDR values <0.05 for their analysis. While this won't change the major thrust of the study, it should give more insight into their pathway analysis.
2. In the text describing the data in Fig 1f, the authors state that "In addition, OCN protein was detected in conditioned media (CM) of human and mouse primary ECs and LRP1 depletion by its specific siRNA dramatically increased its level (Fig. 1f)." LRP1 deletion of HUVEC is not included in the figure. Perhaps this sentence should be re-written to clarify that depletion was only performed in MVEC
3. In several Figures, it was confusing as to which cells were used, and where they came from. For example in Figure 1f, the authors refer to MVEC (mouse microvascular endothelial cells). Are these from lung, liver etc. This should be defined in each Figure legend.
4. Immunoblotting of ocn-promoter driven GFP in various tissues is shown in Figure 2a. Can the authors conclude that expression of GFP is not detected in the heart since the total protein loading (as detected by b-actin) seems low.
5. There is no description how the imaging studies were done (for example, the data in Supp Fig 3c). These data would be significantly enhanced if image analysis were quantify the degree of co-localization between LRP1 and CA-Fox01.
6. The data in Figure 7e demonstrate that during glucose tolerance tests, STZ-injected LRP1 eKO mice display more efficient glucose clearance than WT mice. However, the data in Fig 7d shown that glucose levels in WT mice treated with STZ are significantly higher than KO. Why is this trend not seen in the data of Fig 7e?

Reviewer #2 (Remarks to the Author):

The authors report results that show that depletion of lipoprotein receptor-related protein 1 (LRP1) induces OCN expression in ECs. They show that depletion of OCN abolishes the glucose-lowering effect of LRP1 depletion, OCN treatment normalizes hyperglycemia in a streptozocin HFD mouse model. Several issues need to be addressed:

1. Although the investigators use Cre-driven targeting they need to show a. The % deletion of LRP1 in endothelial cells. They also need to show that LRP1 was not significantly depleted in monocytes which express osteocalcin (refs 37-40). This is important as there is increased monocyte infiltration in insulin sensitive tissues such as liver and skeletal muscle in diabetes.
2. Is the streptozocin plus HFD an appropriate model for type 1 DM or is it a type 2 model?

3. What is the relevance of studying ECs derived from lung and heart in studying insulin sensitivity?
4. What effect does LRP1 deletion have on endothelial function. Does it improve eNOS activation/NO production and increase delivery of insulin to insulin sensitive tissues such as liver and skeletal muscle?
5. What effect does osteocalcin have on IGF-1 and Insulin receptor function in skeletal muscle and liver tissue?
6. When Student t testing was used were the data checked for normative distribution?
7. The suggestion in the discussion that osteocalcin could be used for therapy in type 1 diabetes is a stretch as one has to consider adverse effect such as vascular calcification etc.
8. Why were only males used in these studies.

Reviewer #3 (Remarks to the Author):

In a previous study (Mao et al. Nat Comm 2017), this group has established that endothelial cells (EC) Lrp1 regulates whole body energy metabolism in part by acting as a co-regulator of PPAR-gamma. In the current manuscript, Mao and colleagues focus on the hormone osteocalcin as another potential downstream effector of EC Lrp1. Comparing the transcriptomes of control and Lrp1-deficient EC, they observed that the mRNA of osteocalcin (also called Ocn1/2 or Bglap1/2) was upregulated in absence of Lrp1. Given that OCN, a protein previously characterized as an osteoblast-derived hormone, is known to promote insulin sensitivity, insulin secretion and energy expenditure (Lee et al. Cell 2007, Ferron et al. PNAS 2008), they further investigate if endothelial cells could be a significant source of OCN and if OCN is involved in the metabolic improvement of the Lrp1-EC specific KO mice. Using qPCR on cell cultures of EC and a transgenic mouse expressing GFP under the control of the human OCN promoter, they conclude that EC expressed OCN. Mechanistically, they propose that Lrp1 deletion reduced nuclear FOXO1, a known transcriptional repressor of OCN gene in osteoblasts (Rached et al. JCI 2010). In contrast with previous work by other (Mera et al. Cell Metab 2016; Lin et al. Calcif Tissue Int. 2018) they proposed that OCN regulates insulin signaling by directly interacting with the insulin receptor and IGF1R, thereby modulating glucose uptake by GLUT4. Finally, in the context of streptozotocin-induced diabetes, they show that whole body OCN knockdown using AAV-delivered shRNA blunted the positive effect of Lrp1-deletion in EC on glucose metabolism.

Overall, the observation that EC may regulate whole body glucose metabolism is quite interesting. However, this manuscript suffers from several major issues. Most importantly the vast majority of published data regarding OCN expression in mice and humans indicate that this protein is produced exclusively by osteoblasts and osteocytes. Detailed comments are provided below.

Major comments:

1) Previous works by Desbois et al. (JBC 1994), Sato et al. (BBRC, 1995), Oury et al. (Cell, 2011) have shown that Bglap1 and Bglap2, which encode OCN, are expressed in osteoblasts and in bone, but not in lung, heart and liver, where EC are found. Moreover, publicly available and published single cell RNAseq data (e.g., Tabula Muris) shows that Tie2 expressing cells don't express detectable level of Bglap1/2 in liver, heart and lung. Apart from these contradictory studies, the current manuscript does not provide evidence supporting the notion that EC actually secrete significant amount of OCN protein, except a Western blot (Fig. 1F) using an antibody which was never validated with OCN^{-/-} cells or mice. Measurement of OCN concentration in cell supernatant with a reliable ELISA (i.e., Quidel Inc. Cat # 60-1305) should be performed. The concentration of OCN in the supernatant of osteoblasts and of endothelial cells cultures should be compared side by side to be able to assess how much OCN is actually produced by EC. The same approach (ELISA) should be used to corroborate Figure 3F data.

2) In link with the previous points, qPCR data (e.g., Fig. 1C, E, G) are presented as fold change which does not reveal the absolute level of expression of Bglap1/2 in EC compared to bone and osteoblasts. What is the cycle threshold (Ct) value for the Bglap qPCR in bone vs. EC (Fig. 1G)? Based on previous studies, Bglap1/2 expression in osteoblast and bone should be detected as early as 17 cycles of qPCR. It will be important to show that such level of expression can be detected in

EC. Moreover, it seems that whole bone, instead of marrow flushed bone were used to prepare RNA in figure 1G. This is problematic, as bone marrow has a higher cell density compared to bone tissue.

3) One possible way of interpreting the data presented in Fig. 1D (the increase in circulating bioactive osteocalcin) is that the inactivation of Lrp1 in EC indirectly impacts bone remodeling, leading to a change in circulating level of osteocalcin. A careful histological analysis of bone turnover (i.e., bone formation and bone resorption) should be completed. This should include calcein double labeling to measure bone formation rate (BFR), an index which is often reflected by circulating osteocalcin levels.

4) Similarly, the systemic knockdown of OCN using AAV-shRNA in Figure 7 does not rule out that osteoblasts are responsible for the circulating change in OCN in the eKO of Lrp1. Presumably the AAV also delivered the shRNA to bone cells. This should be acknowledged in the discussion and potentially addressed experimentally.

5) Total OCN in Fig. 1D and 7C appears to have been inferred from the Glu and Gla measurements, which is incorrect. Total OCN should be measured using a specific ELISA (i.e., Quidel Inc. Cat # 60-1305).

6) The rationale for pooling lung and heart EC in the RNA-seq analysis presented in Figure 1A is not clear. In addition, the data appears to be in contradiction with the expression pattern of the hOC-GFP transgene which is not expressed in liver, lung and heart (Figure 2A).

7) The data generated using the hOC-GFP^{tpz} transgenic are interesting, but potentially not physiologically relevant. First, this transgene is driven by a human promoter, which may display ectopic/non-specific expression in mice. These data should be corroborated with direct measurement of endogenous *Bglap1/2* expression using for instance *in situ* hybridization and immunofluorescence. Second, no GFP expression is detected by Western blot in liver, lung and heart, the three tissues from which EC were isolated to perform the RNA seq presented in figure 1A. Third, Figure 1B is lacking appropriate negative controls, i.e., non-transgenic mice.

8) The co-IP experiments presented in figure 3A were generated in HEK 293 cells. These data should be confirmed in EC.

9) In Figure 4 the number of mice analyzed in each group is too small (n=4-5). In panel 4F, insulin seems to be lower in the eKO mice as compare to WT mice before STZ, but higher after STZ. This result suggest that Lrp1 may affect beta cell function or mass differently depending on the metabolic setting (non-diabetic vs. diabetic). How was pancreatic beta cell mass before and after STZ in both genotypes?

10) Fig. 4, Supp. 4 and Supp. 5 are somehow redundant with their previous publication (Mao et al. Nat Comm 2017). Some of the data presented in Figures 5 and 6 are not novel as well. Mera et al. (Cell Metab, 2016) have previously shown that OCN induced AKT phosphorylation, GLUT4 translocation to the membrane and glucose uptake in muscle cell. Other groups have previously reported the beneficial effect of recombinant OCN injection in mice fed HFD (Ferron et al. Bone, 2010; Zhou et al. Endocrinology, 2013).

11) In figure 6 and 7, it is not clear why a GST-OCN fusion protein was use instead of purified recombinant OCN (i.e., without a GST tag). In this setting, the proper control should be GST protein, not saline. In addition, the data of Figure 7E and 7F, which should be presented in the same graph, suggest that OCN knockdown using AAV-shRNA improved glucose tolerance in both control and eKO of Lrp1. This result is not consistent with the data presented in Figure 7I where recombinant OCN reduced glycemia in diabetic mice. This apparent discrepancy should be addressed.

12) A detailed table about the characteristic (age, BMI, sex, blood glucose, etc.) of the human subjects used in Figure 6A is lacking.

13) Proper control groups are often missing. For instance, in Figure 4A-E and 6A-D there is no mice fed control chow (CC), although the text does state that circulating level of osteocalcin are lower in WT mice fed HFD compared to WT fed CC diet. Similarly, in Figure 6F saline injected group is missing. In Figure 7 control (no STZ) WT and eKO mice are missing. Having these controls would be particularly important in Figure 7H to determine if the STZ really decreased the circulating level of insulin.

14) The induction of insulin receptor phosphorylation by OCN is puzzling since Oury et al. (Cell 2011) have previously shown that OCN does not act on cells through a tyrosine kinase receptor.

15) In Figure 5D-F OCN was FLAG tagged. It is not indicated whether the tag is in C- or N-terminus of the protein. In addition, did the authors verify if this OCN-FLAG protein was secreted? Otherwise, the interaction they detected could be an artefact due to the intracellular aggregation of the two overexpressed proteins (OCN and IGF1R or IR). Additional radiolabeled binding assays should be performed to support the conclusion that OCN binds the IR.

Minor comments:

1) The source of the GPRC6A antibody is not indicated in the material and method.

2) What is the genotype of the "WT" mice used in Fig. 1, 4 and 7? If they are flox/flox, did the author verified that the Cdh5-CreER mice do not display any metabolic phenotype and that this Cre line is not ectopically expressed in osteoblasts?

RESPONSE TO THE REVIEWERS

Reviewer #1 (Remarks to the Author):

In the current study, the authors generate an genetic depletion of the LDL receptor-related protein 1 (LRP1) in endothelial cells and identify osteocalcin (OCN) as an upregulated gene in the knockouts. Using a unique ocn promoter-driven report mouse, they confirm expression of this molecule in the vasculature endothelium, and go on to define the molecular pathway by which LRP1 regulates this molecule. Further, they demonstrate that OCN regulates glucose metabolism thereby identifying an important role for the endothelium in regulating insulin sensitivity.

Overall, this is a careful and well written study that includes a variety of approaches and genetically modified mice to support the conclusions reached. There are a few issues that the authors need to address:

1. The selection criteria for the RNA seq data presented in Figure 1a and b and in supplementary Figure 2 are not very stringent (i.e. p values <0.05 , $FC > 1.25$). The authors should correct the p values for false discovery rates and use FDR values <0.05 for their analysis. While this won't change the major thrust of the study, it should give more insight into their pathway analysis.

This is a great comment. As the Reviewer suggested, we re-analyzed the RNA-seq data with more stringent method to correct the p values with false discovery rates (FDRs) <0.05 . The results have been updated in Fig. 1a, b, Supplementary Table 1 and Supplementary Fig. 2. Please note that OCN is still one of most upregulated genes in LRP1 depleted ECs.

2. In the text describing the data in Fig 1f, the authors state that "In addition, OCN protein was detected in conditioned media (CM) of human and mouse primary ECs and LRP1 depletion by its specific siRNA dramatically increased its level (Fig. 1f)." LRP1 deletion of HUVEC is not included in the figure. Perhaps this sentence should be re-written to clarify that depletion was only performed in MLEC cells.

Thank you very much for your comments. As the Reviewer suggested, we updated the manuscript as shown here, "In addition, OCN protein was detected in conditioned media (CM) of human and mouse primary ECs and LRP1 depletion in MHLECs by its specific siRNA dramatically increased its level (Fig. 1f)".

3. In several Figures, it was confusing as to which cells were used, and where they came from. For example in Figure 1f, the authors refer to MVEC (mouse microvascular endothelial cells). Are these from lung, liver etc. This should be defined in each Figure legend.

We apologize for this confusing point. To clarify this issue, we included the specific information in the related figure legends. In Fig. 1f, MVEC (mouse microvascular endothelial cells) were from the lung. So we labeled them as MLECs.

4. Immunoblotting of ocn-promoter driven GFP in various tissues is shown in Figure 2a. Can the authors conclude that expression of GFP is not detected in the heart since the total protein loading (as detected by b-actin) seems low.

We appreciate the Reviewer's comments. By taking longer exposure time, we detected fair amount of beta actin in the heart (revised Fig. 2a). However, the expression of GFP was still very low. Therefore, we conclude that human *ocn* promoter-driven GFP expression is mainly enriched in aorta, skeletal muscle and eye besides of bones.

5. There is no description how the imaging studies were done (for example, the data in Supp Fig 3c). These data would be significantly enhanced if image analysis were used to quantify the degree of co-localization between LRP1 and CA-FoxO1.

Answer: We apologize for this oversight. A sub-section “**Immunofluorescent studies**” has been included in the Methods section. In addition, the co-localization between LRP1 and CA-FoxO1 has been quantitatively analyzed with Coloc 2 plug-in of the Fiji software and results have been included in Supplementary Fig. 3c.

6. The data in Figure 7e demonstrate that during glucose tolerance tests, STZ-injected LRP1 eKO mice display more efficient glucose clearance than WT mice. However, the data in Fig 7d show that glucose levels in WT mice treated with STZ are significantly higher than KO. Why is this trend not seen in the data of Fig 7e?

This is a great question. Glucose levels shown in Fig. 7d were measured with non-fasting mouse serum. To study glucose tolerance, mice were fasted for 6 hours before glucose challenge which diminished the differences between WT and eKO mice (Fig. 7e). The figure legend and methods have been updated in order to clarify this point.

Reviewer #2 (Remarks to the Author):

The authors report results that show that depletion of lipoprotein receptor-related protein 1 (LRP1) induces OCN expression in ECs. They show that depletion of OCN abolishes the glucose-lowering effect of LRP1 depletion, OCN treatment normalizes hyperglycemia in a streptozocin HFD mouse model.

Several issues need to be addressed:

1. Although the investigators use Cre-driven targeting they need to show a. The % deletion of LRP1 in endothelial cells. They also need to show that LRP1 was not significantly depleted in monocytes which express osteocalcin (refs 37-40). This is important as there is increased monocyte infiltration in insulin sensitive tissues such as liver and skeletal muscle in diabetes.

This is a great suggestion. We isolated monocytes and endothelial cells and measured LRP1 mRNA levels using real-time PCR assays. As expected, there was no decrease in LRP1 mRNA level in monocytes (Supplementary Fig. 1b). Together with non-reduction of LRP1 levels in leukocytes, liver, skeletal muscle and WAT tissues (Supplementary Fig. 1a), it suggests that LRP1 was specifically depleted in ECs of LRP1 eKO mice and that osteocalcin induction is not contributed by monocytes of LRP1 eKO mice.

2. Is the streptozocin plus HFD an appropriate model for type 1 DM or is it a type 2 model?

In this study, we used two mouse models, one is the streptozocin-induced diabetic mouse as a type 1 diabetes model (Fig. 4f-m, 7). In addition, HFD-fed mice were used to study type 2 diabetes (Fig. 4a-e, 6). We did not treat mice with both streptozocin and HFD in this study. We hope this explanation could clarify the confusion.

3. What is the relevance of studying ECs derived from lung and heart in studying insulin sensitivity?

We totally understand the reviewer's concern that ECs isolated from different vessel beds might behave differently and agree that the perfect way would be to study ECs with targeted metabolic organs. However,

technical difficulty still exists for isolating primary ECs from a variety of mouse tissues. In order to culture several passages of ECs, we need isolate from very young mice (<2 weeks old). Therefore, isolated pure EC population are limited due to the low abundance of many tissues. Till now, a reproducible and well-tested method has been established for the culture of primary ECs isolated from mouse lung and heart¹. Our lab has been using this method to generate highly pure and functionally competent ECs for several years. Although we could isolate and culture ECs from the liver, liver ECs were not studied for this project since liver ECs are more heterogenous due to the unique sinusoidal ECs that might compound our data interpretation.

Since one important function of ECs in the regulation of insulin sensitivity is through the activation of eNOS by insulin, we tested whether ECs isolated from lung were responsive to insulin. As expected, insulin increased eNOS phosphorylation, suggesting the activation of eNOS (Rebuttal Fig. 1). These data suggest that MLECs could be used to study the underlying mechanisms for EC regulation of insulin sensitivity.

4. What effect does LRP1 deletion have on endothelial function. Does it improve eNOS activation/NO production and increase delivery of insulin to insulin sensitive tissues such as liver and skeletal muscle?

This is a very interesting question! We studied how LRP1 depletion impacts insulin-induced eNOS activation in MLECs. As expected, insulin increased eNOS phosphorylation, indicating the activation of eNOS (Rebuttal Fig. 1). Interestingly, LRP1 knockdown led to a marked increase of eNOS phosphorylation at the basal condition and a very mild increase in response to insulin (Rebuttal Fig. 1). It suggests that LRP1 depletion promotes eNOS activity in ECs. Given that eNOS activation/NO production is critical for insulin delivery to liver, skeletal muscle and other insulin-sensitive tissues², we hypothesize that LRP1 regulation of eNOS activity might also contribute to the improvement of insulin sensitivity in LRP1 eKO mice. Therefore, this point has been included in the Discussion section, as shown as following, “Our previous studies demonstrate critical roles of LRP1 in endothelial cell function, such as angiogenesis, inflammation and lipid transport³⁻⁶. These processes regulated by LRP1 might also contribute to the improvement of glucose metabolism in LRP1 eKO mice, which warrants further investigation.”.

5. What affect does osteocalcin have on IGF-1 and Insulin receptor function in skeletal muscle and liver tissue?

To investigate how OCN impacts IGF-1 and insulin receptor function in metabolic tissues, we injected mice with OCN and measured the phosphorylatory status of these receptors. In skeletal muscle and liver, OCN increased phosphorylation of IR and IGF1R at tyrosine 1131 and 1146, respectively (Fig. 5a-b, Supplementary Fig. 6a-b. In addition, IGF1R knockdown inhibited OCN-activated insulin downstream signaling in hepatocytes and glucose uptake in C2C12 cells (Fig. 5h-i). Taken together, our results suggest that OCN promotes the activation of IR and IGF1R and their dependent glucose handling processes.

6. When Student t testing was used were the data checked for normative distribution?

Answer: Yes, we performed “Normality and lognormality Tests” for the datasets. Student *t*-tests were used for two-group sample comparison when these data passed for the normality of distribution. Otherwise, Mann-Whitney *U* tests were used for their comparison. The Quantification and Statistical Analysis” section has been updated with more details accordingly.

7. *The suggestion in the discussion that osteocalcin could be used for therapy in type 1 diabetes is a stretch as one has to consider adverse affect such as vascular calcification etc.*

This is a great comment. We have modified these sentences in the Discussion section, as shown as following, “Our studies suggest that OCN could improve glucose tolerance in both insulin-deficient and insulin-resistant conditions. However, adverse effects might exist for OCN since it is often associated with vascular calcification^{7,8}. Therefore, understanding the underlying mechanisms of OCN regulation of glucose metabolism and dissecting its interplay with insulin and IGF1 will shed light on how to use it to treat diabetes in an effective and safe manner.”.

8. *Why were only males used in these studies.*

As we know, males and females differ in many steps of nutrient handling including adipose triglyceride storage and lipolysis, and also liver fatty acid uptake, triglyceride synthesis, VLDL biology, cholesterol uptake and HDL function. Female sex affords protection against coronary heart disease and diabetes, with some studies suggesting females are at half the risk^{9,10}. Consistently, female C57BL/6J mice are relative resistance to diet-induced obesity than male mice^{11,12}. In order to exclude the gender interference, we only included male mice in this study. The investigation to understand the role of EC-LRP1 depletion in female mice is considered as one of our future research focuses.

Reviewer #3 (Remarks to the Author):

In a previous study (Mao et al. Nat Comm 2017), this group has established that endothelial cells (EC) Lrp1 regulates whole body energy metabolism in part by acting as a co-regulator of PPAR-gamma. In the current manuscript, Mao and colleagues focuses on the hormone osteocalcin as another potential downstream effector of EC Lrp1. Comparing the transcriptomes of control and Lrp1-deficient EC, they observed that the mRNA of osteocalcin (also called Ocn1/2 or Bglap1/2) was upregulated in absence of Lrp1. Given that OCN, a protein previously characterized as an osteoblast-derived hormone, is known to promote insulin sensitivity, insulin secretion and energy expenditure (Lee et al. Cell 2007, Ferron et al. PNAS 2008), they further investigate if endothelial cells could be a significant source of OCN and if OCN is involved in the metabolic improvement of the Lrp1-EC specific KO mice. Using qPCR on cell cultures of EC and a transgenic mouse expressing GFP under the control of the human OCN promoter, they conclude that EC expressed OCN. Mechanistically, they propose that Lrp1 deletion reduced nuclear FOXO1, a known transcriptional repressor of OCN gene in osteoblasts (Rached et al. JCI 2010). In contrast with previous work by other (Mera et al. Cell Metab 2016; Lin et al. Calcif Tissue Int. 2018) they proposed that OCN regulates insulin signaling by directly interacting with the insulin receptor and IGF1R, thereby modulating glucose uptake by GLUT4. Finally, in the context the streptozotocin-induced diabetes, they show that whole body OCN knockdown using AAV-delivered shRNA blunted the positive effect of Lrp1-deletion in EC on glucose metabolism.

Overall, the observation that EC may regulate whole body glucose metabolism is quite interesting. However, this manuscript suffers from several major issues. Most importantly the vast majority of published data regarding OCN expression in mice and humans indicate that this protein is produced exclusively by osteoblasts and osteocytes. Detailed comments are provided below.

Major comments:

1) Previous works by Desbois et al. (JBC 1994), Sato et al. (BBRC, 1995), Oury et al. (Cell, 2011) have

shown that Bglap1 and Bglap2, which encode OCN, are expressed in osteoblasts and in bone, but not in lung, heart and liver, where EC are found. Moreover, publicly available and published single cell RNAseq data (e.g., Tabula Muris) shows that Tie2 expressing cells don't express detectable level of Bglap1/2 in liver, heart and lung. Apart from these contradictory studies, the current manuscript does not provide evidence supporting the notion that EC actually secrete significant amount of OCN protein, except a Western blot (Fig. 1F) using an antibody which was never validated with OCN^{-/-} cells or mice. Measurement of OCN concentration in cell supernatant with a reliable ELISA (i.e., Quidel Inc. Cat # 60-1305) should be performed. The concentration of OCN in the supernatant of osteoblasts and of endothelial cells cultures should be compared side by side to be able to assess how much OCN is actually produced by EC. The same approach (ELISA) should be used to corroborate Figure 3F data.

We totally understand the Reviewer's concerns and appreciate the great suggestion for a reliable OCN ELISA experiment. Therefore, we performed additional experiments to compare the secreted OCN levels in conditioned media collected from osteoblasts and mouse lung ECs using the Quidel OCN ELISA kit. The OCN level in osteoblast (Ob)-derived conditioned media (CM) were ~5-fold higher than that in EC-derived CM (5.84±0.31 ng/mg protein in Ob-CM compared to 1.23±0.23 ng/mg protein in EC-CM, Fig. 1h). This OCN ELISA assay has been calibrated with OCN standard curve (data not shown) and validated with OCN knockdown ECs where OCN level was decreased in their CM (from 1.70±0.19 ng/mg protein to 0.21±0.04 ng/mg protein, Supplementary Fig. 2e). These results have been added into the Results section accordingly.

2) In link with the previous points, qPCR data (e.g., Fig. 1C, E, G) are presented as fold change which does not reveal the absolute level of expression of Bglap1/2 in EC compared to bone and osteoblasts. What is the cycle threshold (Ct) value for the Bglap qPCR in bone vs. EC (Fig. 1G)? Based on previous studies, Bglap1/2 expression in osteoblast and bone should be detected as early as 17 cycles of qPCR. It will be important to show that such level of expression can be detected in EC. Moreover, it seems that whole bone, instead of marrow flushed bone were used to prepare RNA in figure 1G. This is problematic, as bone marrow has a higher cell density compared to bone tissue.

Regarding to the qPCR data, we used Roche Universal ProbeLibrary probe/primer sets for the real-time PCR assays. Gene expression changes were analyzed through subtracting these genes' Ct values by the Ct values of housekeeping genes (i.e. *gapdh*, *beta-actin*), a well-established relative quantitative methods. In Fig. 1C, the Ct values for OCN were ~35 in control ECs and 31~32 in LRP1 knockdown ECs. In Fig. 1E, for OCN1, the Ct values were ~35 in control ECs and 33 in LRP1 knockdown ECs, and for OCN2, the Ct values were 38 in control ECs and 34 in LRP1 knockdown ECs. In Fig. 1G, the Ct values were 34 for WT ECs and 30 for eKO ECs, 39 for both WT and eKO bones and 33 for osteoblasts. After normalizing to the Ct values of GAPDH in ECs, bones and osteoblasts, OCN level was ~3.0-fold and 6.2-fold lower in ECs than its level in bones and osteoblasts. Our Ct values for osteoblast and bone are different from that Ct at 17 mentioned by the Reviewer, which is likely due to the different amounts of cDNAs used for qPCRs and sensitivity of probe and primers in PCR reaction mixes.

We would also like to apologize for the confusion about what bone samples we were used for our studies. Indeed, we flushed bones with PBS to remove bone marrow before performing OCN and LRP1 expression assays. The method has been revised accordingly.

Taken together, our results based on qPCR, Western blotting and ELISA suggest that OCN is expressed in ECs and its expression is regulated by LRP1.

3) One possible way of interpreting the data presented in Fig. 1D (the increase in circulating bioactive osteocalcin) is that the inactivation of Lrp1 in EC indirectly impacts bone remodeling, leading to a change

in circulating level of osteocalcin. A careful histological analysis of bone turnover (i.e., bone formation and bone resorption) should be completed. This should include calcein double labeling to measure bone formation rate (BFR), an index which is often reflected by circulating osteocalcin levels.

This is a very thoughtful comment. As reviewer suggested, we performed calcein double labeling experiment to evaluate bone formation rate (BFR) in LRP1 eKO and WT mice^{13,14}. As shown in Supplementary Fig. 2f, the BFR of LRP1 eKO mice were not significantly different from their littermate control WT mice. These data suggest that EC-specific LRP1 depletion did not significantly impact on bone turnover in mice.

4) Similarly, the systemic knockdown of OCN using AAV-shRNA in Figure 7 does not rule out that osteoblasts are responsible for the circulating change in OCN in the eKO of Lrp1. Presumably the AAV also delivered the shRNA to bone cells. This should be acknowledged in the discussion and potentially addressed experimentally.

We totally agree with the reviewer's comments. Previous paper showed OCN is an osteoblast-secreted metabolic hormone. In our paper, we discovered that the endothelium could be another important source for secret OCN besides of osteoblasts. AAV-delivered OCN shRNA could lead to OCN knockdown in a variety of cell type, including bone and ECs. Therefore, the "rescue" effect for AAV-OCN shRNA could be maximized due to OCN knockdown in a variety of cells, including ECs and osteoblasts. We have included more discussion in the manuscript, "Our data suggest that vascular endothelium could be another important source for OCN. In our studies with AAV-OCN shRNA injection, OCN depletion was not limited to ECs. Therefore, the "rescue" effect for AAV-OCN shRNA could be maximized due to OCN knockdown in a variety of cells (i.e. ECs, osteoblasts).".

5) Total OCN in Fig. 1D and 7C appears to have been inferred from the Glu and Gla measurements, which is incorrect. Total OCN should be measured using a specific ELISA (i.e., Quidel Inc. Cat # 60-1305).

As the Reviewer suggested, the level of total OCN has been measured with the ELSIA kit purchased from Quidel. Similar results have been obtained and the graphs in Fig. 1d and 7c have been updated accordingly.

6) The rationale for pooling lung and heart EC in the RNA-seq analysis presented in Fig. 1A is not clear. In addition, the data appears to be in contradiction with the expression pattern of the hOC-GFP transgene which is not expressed in liver, lung and heart (Figure 2A).

We totally agree that it would be the best to separate the RNA-seq studies for lung and heart ECs. At the current stage of endothelial cell studies, isolation and culture of murine primary ECs from different vessel beds are still difficult. Recently, many laboratories including ours have established a reproducible method to isolate ECs from lung and heart¹. To improve the yield and purity, it is still acceptable to combine lung and heart for microvascular ECs for mechanistic studies. By comparing our isolated lung and heart ECs with commercially available cells, many gene expression changes in heart ECs are very similar as that in the lung. In addition, due to the high cost of the RNA-seq assays that were performed in 2017, we decided to study the pool of microvascular ECs isolated from the heart and lung. After we obtained RNA-seq results, the expression changes have been further confirmed with mouse liver ECs, mouse heart and lung ECs and mouse lung ECs (Fig. 1c, e-h). Taken together, we hypothesize that LRP1 regulates OCN expression in ECs.

As the Reviewer commented, the expression pattern of human OCN is different from mouse OCN. Different from one human *ocn* gene, a gene cluster containing *ocn1* (*ocn*), *ocn2* and *ocn*-related gene (*org*) exists in mouse genome¹⁵. OCN(1) and OCN2 proteins only differ in two amino acids within their signal peptides, while ORG is more different from them. These differences between mouse and human OCN promoter

sequences suggest their expression might be regulated differently. However, LRP1 depletion increased OCN expression in mice and also human *Ocn* promoter-driven GFP expression in a similar manner (Fig. 1c-h for mouse OCN vs Fig. 2f for human OCN). In addition, hyperglycemia downregulated OCN level in mice, GFP signals in human hOC-GFP mice and serum of metabolic syndrome patients and mouse models (Fig. 2e, 7a-c, 6a-d). These data suggest that OCN regulation by LRP1 and hyperglycemia is similar in mouse and human. It raises a need to further characterize the regulatory machinery for human and mouse OCN expression and what controls the differential expression of OCN in human versus mouse and among a variety of tissues. These questions will become one of our future research focuses. Additional discussion regarding this point has been included in the Discussion section.

7) The data generated using the hOC-GFPtpz transgenic are interesting, but potentially not physiologically relevant. First, this transgene is driven by a human promoter, which may display ectopic/non-specific expression in mice. These data should be corroborated with direct measurement of endogenous Bglap1/2 expression using for instance in situ hybridization and immunofluorescence. Second, no GFP expression is detected by Western blot in liver, lung and heart, the three tissues from which EC were isolated to perform the RNA seq presented in figure 1A. Third, Figure 1B is lacking appropriate negative controls, i.e., non-transgenic mice.

Answer: We totally agree with the Reviewer that GFP signals accumulated in cells could be affected by non-specific factors such as half-life and stability of GFP itself besides of OCN promoter-driven expression. However, the hOC-GFPtpz reporter mouse model has been validated as a great tool for the understanding of human OCN promoter activation^{16,17}. In addition, our observations in hOC-GFP transgenic mice (Fig. 2e-f) suggest the regulation of GFP expression by hyperglycemia and LRP1 depletion is similar as that of mouse OCN. OCN protein was also detected in human ECs (human umbilical vein ECs, Fig. 1f). Therefore, we hypothesize that OCN is expressed in ECs and hOC-GFPtpz mice could be at least used as a supportive tool for the study of OCN expression in ECs. As we know, differences exist between human and mouse *ocn* genes. For example, there is one *ocn* gene in human, whereas a gene cluster containing *ocn1* (*ocn*), *ocn2* and *ocn*-related gene (*org*) exists in mouse genome¹⁵. In this study, OCN mRNA and protein were detected in mouse lung, heart and liver but human *Ocn* promoter-driven GFP signals were not detected in these organs. It suggests that OCN could be expressed in ECs in both human and mice, however, as we mentioned in the answer for the #6 question of this Reviewer, the expression pattern of human OCN might be different from mouse OCN. The underlying regulatory mechanisms for the difference between human and mouse OCN expression, including their different tissue distribution, remain to be further studied. We have added more discussion regarding this point in the Discussion section accordingly.

In addition, negative control images with non-transgenic tissue sections were included in Supplementary Fig. 2g.

8) The co-IP experiments presented in figure 3A were generated in HEK 293 cells. These data should be confirmed in EC.

As the Reviewer suggested, we performed IP experiments with anti-LRP1 antibody to enrich its interacting proteins in MLECs. As shown in revised Fig. 3b, endogenous FoxO1, FoxO3a and FoxO4 were detected in the same complex with endogenous LRP1. It suggests that LRP1 could interact with FoxOs in ECs. We have updated the Results section accordingly.

9) In Figure 4 the number of mice analyzed in each group is too small (n=4-5). In panel 4F, insulin seems to be lower in the eKO mice as compare to WT mice before STZ, but higher after STZ. This result suggest that Lrp1 may affect beta cell function or mass differently depending on the metabolic setting (non-diabetic

vs. diabetic). How was pancreatic beta cell mass before and after STZ in both genotypes?

As the Reviewer suggested, we added more mice in these studies and similar results have been obtained. Please see the revised Fig. 4b-e and Supplementary Fig. 5g for HFD-fed mice and Fig. 4i-m for STZ-treated mice.

In addition, we evaluated the beta cell mass differences in LRP1 eKO and WT mice before and after STZ injection. As expected, STZ treatment significantly decreased beta-cell masses in both LRP1 eKO and WT mice compared to their non-STZ controls (Supplementary Fig. 5h). However, no significant beta-cell mass differences were detected between LRP1 eKO and WT mice before or after STZ treatments (Supplementary Fig. 5h). It suggests that this regulation of insulin level in LRP1 eKO mice (Fig. 4f) is not mediated through impacting beta cell function. The underlying mechanisms still need further investigation. These results have been added in the Results section accordingly.

10) Fig. 4, Supp. 4 and Supp. 5 are somehow redundant with their previous publication (Mao et al. Nat Comm 2017). Some of the data presented in Figures 5 and 6 are not novel as well. Mera et al. (Cell Metab, 2016) have previously shown that OCN induced AKT phosphorylation, GLUT4 translocation to the membrane and glucose uptake in muscle cell. Other groups have previously reported the beneficial effect of recombinant OCN injection in mice fed HFD (Ferron et al. Bone, 2010; Zhou et al. Endocrinology, 2013).

We appreciate your great comments. This study stemmed from our previous data that LRP1 depletion in ECs improved insulin sensitivity and glucose homeostasis⁴. In the previous study, the loss-of-function study for LRP1 was performed with LRP1^{fl/fl}; Tie2Cre^{+/+} (Cre+) mice, in which the Lrp1 gene is specifically deleted in ECs and bone marrow-derived hematopoietic cells. Then we transplanted wildtype bone marrow to these mice to restore hematopoietic LRP1 in order to obtain EC-specific knockout mice for LRP1. Although bone marrow-transplanted mice were broadly studied in this field, the radiation might cause some unknown effects. In addition, Tie2-mediated vascular defects during embryonic stage could also impact metabolic dysregulation indirectly. Therefore, to clarify the specific effects of EC-LRP1 depletion during adulthood, in this follow-up study, we used LRP1^{fllox/fllox} and Cdh5-CreER^{+/+} mice to generate the LRP1^{fl/fl}; Cdh5-CreER^{+/+} (WT or eKO) mice, which specifically depleted LRP1 in ECs upon tamoxifen induction. These data provide crucial evidence to further support that EC-LRP1 depletion plays a protective role in glucose homeostasis. More importantly, we identified a new mediator-OCN as a important player for endothelial regulation of glucose homeostasis by LRP1.

Mera et al. (Cell Metab, 2016) showed OCN induced AKT phosphorylation in muscles¹⁸. In this paper, we performed detailed experiments with skeletal muscle and liver samples and observed that phosphorylation of IR, IGF1R, the more upstream players of IR pathway than AKT was also increased by OCN. Together with the immunoprecipitation of insulin receptor complexes and GPRC6A knockdown studies, it suggests that OCN might transactivate IGF/IR through GPRC6A. This study provides further insights for OCN regulation of insulin sensitivity through promoting insulin signaling in metabolic cells.

Mera et al. showed that OCN increases membrane translocation of GLUT4 in skeletal muscle¹⁸. In our study, we observed that OCN, similar as insulin and IGF1, increased membrane translocation of GLUT4 in skeletal muscle. Since these results were somewhat similar as that in the previous reports¹⁸, we moved this figure (Fig. 5c) to Supplementary information section as Supplementary Fig. 6c and Mera's paper has been cited in our manuscript.

Regarding to the glucose uptake study in muscle cells, previous reports¹⁸ suggest that OCN could regulate glucose uptake in muscle cells. In this paper, we used OCN-induced glucose uptake served as a positive

control. Our major finding is that OCN-induced glucose uptake could be blocked by IGF1R depletion (Fig. 5i), suggesting that OCN-regulated glucose handling is mediated through IGF1R pathway.

11) In figure 6 and 7, it is not clear why a GST-OCN fusion protein was used instead of purified recombinant OCN (i.e., without a GST tag). In this setting, the proper control should be GST protein, not saline. In addition, the data of Figure 7E and 7F, which should be presented in the same graph, suggest that OCN knockdown using AAV-shRNA improved glucose tolerance in both control and eKO of Lrp1. This result is not consistent with the data presented in Figure 7I where recombinant OCN reduced glycemia in diabetic mice. This apparent discrepancy should be addressed.

As the Reviewer mentioned, we used purified GST-OCN fusion protein for *in vivo* studies in Fig. 7. The control mice were injected with saline containing comparable amount of GST protein. The label has been updated accordingly.

When performing GTT experiments (Figure 7e, 7f), we could only handle maximally two groups of mice (n=11) at the same time due to the tight schedule for glucose measuring time points. Therefore, we presented their data (Fig. 7e and 7f) separately. Based on GTT area under curve (AUC) analysis, AAV-shRNA did not significantly improve glucose tolerance in either LRP1 WT or eKO mice (Fig. 7g). Therefore, this data is not contradictory to glycemia-lowering effect of OCN (Fig. 7i). On the other hand, STZ-injected LRP1 eKO mice displayed more efficient glucose clearance than WT mice (Fig. 7g). However, this improvement was abolished in OCN AAV-shRNA-injected LRP1 eKO mice (Fig. 7g). Taken together, our results suggest OCN is required for EC-LRP1 depletion to protect mice from T1DM.

12) A detailed table about the characteristic (age, BMI, sex, blood glucose, etc.) of the human subjects used in Figure 6A is lacking.

The baseline characteristics (i.e. age, BMI, sex, blood glucose) of these participants have been listed in the Table 1 in the previously published paper¹⁹. The reference has been cited in the Methods section.

13) Proper control groups are often missing. For instance, in Figure 4A-E and 6A-D there is no mice fed control chow (CC), although the text does state that circulating level of osteocalcin are lower in WT mice fed HFD compared to WT fed CC diet. Similarly, in Figure 6F saline injected group is missing. In Figure 7 control (no STZ) WT and eKO mice are missing. Having these controls would be particularly important in Figure 7H to determine if the STZ really decreased the circulating level of insulin.

We apologize for the confusion. The glucose studies for mice fed HFD are listed in Fig. 4A-E, while that for mice fed control chow are in Supplementary Fig. 4. In addition, the circulating levels of osteocalcin are listed in Fig. 6b-d for mice fed HFD and in Fig. 7a-c for STZ-treated mice, while that for control (non-STZ) mice fed CC are in Fig. 1d. Also, the saline group was added in Fig. 6f and the control WT and eKO groups for Fig. 7h are shown in Supplementary Fig. 5b. To clarify these points, the figure legends for these figures have been updated accordingly.

14) The induction of insulin receptor phosphorylation by OCN is puzzling since Oury et al. (Cell 2011) have previously shown that OCN does not act on cells through a tyrosine kinase receptor.

Answer: We were surprised too when we initially detected the phosphorylation of insulin receptor upon OCN treatment. To understand the underlying mechanisms, we investigated whether OCN could form a protein complex with IGF1R and IR. Our immunoprecipitation studies demonstrated that OCN could form a complex with IGF1R and IR, and IGF1R also formed a complex with OCN binding protein-GPRC6A (Fig. 5c-g). In addition, GPRC6A knockdown by its specific siRNAs inhibited OCN-promoted IRS1 phosphorylation. Therefore, we speculate that OCN might transactivate IR/IGF1R signaling through

GPRC6A. The detailed mechanism by which IR/IGF1R activation is promoted by OCN/GPRC6A still needs further investigation, which will become one of our future plans.

15) In Figure 5D-F OCN was FLAG tagged. It is not indicated whether the tag is in C- or N-terminus of the protein. In addition, did the authors verify if this OCN-FLAG protein was secreted? Otherwise, the interaction they detected could be an artefact due to the intracellular aggregation of the two overexpressed proteins (OCN and IGF1R or IR). Additional radiolabeled binding assays should be performed to support the conclusion that OCN binds the IR.

OCN was tagged with Flag epitope at its c-terminus. To produce Flag-tagged OCN protein, HEK293 cells were transfected with Flag-OCN. Then, OCN protein was enriched through immunoprecipitation with anti-Flag antibody-conjugated beads followed by elution with Flag peptide and passing through protein filter with cut-off size at 10 kDa. In Fig. 5c-f, HEK293 cells were transfected with V5-tagged IGF1R or GFP-tagged IR and then treated with Flag-OCN. In Fig. 5g, primary hepatocytes were treated with Flag-OCN. To stabilize the formed protein complex of OCN, cells were cross-linked with a reversible crosslinker-DSP (dithiobis(succinimidyl propionate)). Through immunoprecipitation with anti-Flag antibody, we detected overexpressed or endogenous IGF1R and IR in OCN-enriched protein complex. Taken together, we hypothesize that OCN could form a protein complex with IGF1R and IR. These results and related methods have been updated in the Results and Methods sections accordingly.

Minor comments:

1) The source of the GPRC6A antibody is not indicated in the material and method.

Apologize for this oversight. The GPRC6A antibody was purchased from Sigma. This information has been included in the Supplemental Information section.

2) What is the genotype of the “WT” mice used in Fig. 1, 4 and 7? If they are flox/flox, did the author verified that the Cdh5-CreER mice do not display any metabolic phenotype and that this Cre line is not ectopically expressed in osteoblasts?

Our WT mice used in Fig. 1, 4 and 7 were LRP1^{flox/flox};Cdh5-CreER^{-/-} mice, which have been indicated in the text and figure legend. Cdh5-Cre-mediated knockout mouse models for genes (i.e. flt1²⁰, insulin receptor²¹ and argonaute 1²²) have been studied for metabolic phenotype by many laboratories. Neither we nor other groups have detected metabolic phenotypes with Cdh5-Cre or Cdh5-CreER transgenic mice.

The specific expression of Cdh5 gene in vascular endothelial cells has been well characterized with multiple mouse models, such as its reporter mice and loss-of-function mice²³⁻²⁶. Particularly, its expression is limited to vascular endothelial cells in the bones during the embryonic stage and adulthood^{23,24}. Recent studies with single cell-RNA sequencing analysis further demonstrate that Cdh5-positive ECs and osteoblasts are two distinct groups in the bones²⁷. We also tested the expression of Cdh5 expression in the cultured osteoblasts and no signals were detected using real-time PCR assays (data not shown). Taken together, Cdh5-cre is not active in osteoblasts.

References

1. Lim, Y.C. & Lusinskas, F.W. Isolation and culture of murine heart and lung endothelial cells for in vitro model systems. *Methods Mol Biol* **341**, 141-154 (2006).
2. Pi, X., Xie, L. & Patterson, C. Emerging roles of vascular endothelium in metabolic homeostasis. *Circ Res* **123**, 477-494 (2018).
3. Lockyer, P., *et al.* LRP1-dependent BMPER signaling regulates lipopolysaccharide-induced vascular inflammation. *Arterioscler Thromb Vasc Biol* **37**, 1524-1535 (2017).

4. Mao, H., *et al.* Endothelial LRP1 regulates metabolic responses by acting as a co-activator of PPARgamma. *Nat Commun* **8**, 14960 (2017).
5. Mao, H., Lockyer, P., Townley-Tilson, W.H., Xie, L. & Pi, X. LRP1 regulates retinal angiogenesis by inhibiting PARP-1 activity and endothelial cell proliferation. *Arterioscler Thromb Vasc Biol* **36**, 350-360 (2016).
6. Pi, X., *et al.* LRP1-dependent endocytic mechanism governs the signaling output of the bmp system in endothelial cells and in angiogenesis. *Circ Res* **111**, 564-574 (2012).
7. Idelevich, A., Rais, Y. & Monsonego-Ornan, E. Bone Gla protein increases HIF-1alpha-dependent glucose metabolism and induces cartilage and vascular calcification. *Arterioscler Thromb Vasc Biol* **31**, e55-71 (2011).
8. Tacey, A., *et al.* Potential role for osteocalcin in the development of atherosclerosis and blood vessel disease. *Nutrients* **10**(2018).
9. Lloyd-Jones, D., *et al.* Heart disease and stroke statistics--2009 update: a report from the American Heart Association Statistics Committee and Stroke Statistics Subcommittee. *Circulation* **119**, 480-486 (2009).
10. Regensteiner, J.G., *et al.* Sex differences in the cardiovascular consequences of diabetes mellitus: a scientific statement from the American Heart Association. *Circulation* **132**, 2424-2447 (2015).
11. Pettersson, U.S., Walden, T.B., Carlsson, P.O., Jansson, L. & Phillipson, M. Female mice are protected against high-fat diet induced metabolic syndrome and increase the regulatory T cell population in adipose tissue. *PLoS One* **7**, e46057 (2012).
12. Hong, J., Stubbins, R.E., Smith, R.R., Harvey, A.E. & Nunez, N.P. Differential susceptibility to obesity between male, female and ovariectomized female mice. *Nutr J* **8**, 11 (2009).
13. Kim, H.J., *et al.* Glucocorticoids suppress bone formation via the osteoclast. *J Clin Invest* **116**, 2152-2160 (2006).
14. Porter, A., *et al.* Quick and inexpensive paraffin-embedding method for dynamic bone formation analyses. *Sci Rep* **7**, 42505 (2017).
15. Desbois, C., Hogue, D.A. & Karsenty, G. The mouse osteocalcin gene cluster contains three genes with two separate spatial and temporal patterns of expression. *J Biol Chem* **269**, 1183-1190 (1994).
16. Bilic-Curcic, I., *et al.* Visualizing levels of osteoblast differentiation by a two-color promoter-GFP strategy: Type I collagen-GFPcyan and osteocalcin-GFPtpz. *Genesis* **43**, 87-98 (2005).
17. Ushiku, C., Adams, D.J., Jiang, X., Wang, L. & Rowe, D.W. Long bone fracture repair in mice harboring GFP reporters for cells within the osteoblastic lineage. *J Orthop Res* **28**, 1338-1347 (2010).
18. Mera, P., *et al.* Osteocalcin signaling in myofibers is necessary and sufficient for optimum adaptation to exercise. *Cell Metab* **23**, 1078-1092 (2016).
19. Khan, I.M., *et al.* Postprandial monocyte activation in individuals with metabolic syndrome. *J Clin Endocrinol Metab* **101**, 4195-4204 (2016).
20. Seki, T., *et al.* Ablation of endothelial VEGFR1 improves metabolic dysfunction by inducing adipose tissue browning. *J Exp Med* **215**, 611-626 (2018).
21. Wang, X., *et al.* Insulin resistance in vascular endothelial cells promotes intestinal tumour formation. *Oncogene* **36**, 4987-4996 (2017).
22. Tang, X., *et al.* Suppression of endothelial AGO1 promotes adipose tissue browning and improves metabolic dysfunction. *Circulation* **142**, 365-379 (2020).
23. Monvoisin, A., *et al.* VE-cadherin-CreERT2 transgenic mouse: a model for inducible recombination in the endothelium. *Dev Dyn* **235**, 3413-3422 (2006).

24. Alva, J.A., *et al.* VE-Cadherin-Cre-recombinase transgenic mouse: a tool for lineage analysis and gene deletion in endothelial cells. *Dev Dyn* **235**, 759-767 (2006).
25. Pu, W., *et al.* Genetic targeting of organ-specific blood vessels. *Circ Res* **123**, 86-99 (2018).
26. Chen, Q., *et al.* Apelin(+) endothelial niche cells control hematopoiesis and mediate vascular regeneration after myeloablative injury. *Cell Stem Cell* **25**, 768-783 e766 (2019).
27. Baryawno, N., *et al.* A cellular taxonomy of the bone marrow stroma in homeostasis and leukemia. *Cell* **177**, 1915-1932 e1916 (2019).

Reviewer comments, second round of review:

Reviewer #1 (Remarks to the Author):

In the revised manuscript, the authors have addressed all of my concerns. I feel that this is an important study connecting the endothelium, LRP1, osteocalcin and glucose metabolism.

Reviewer #2 (Remarks to the Author):

The authors have answered most questions of reviewers. My only suggestion at this stage is acknowledge the use of lung endothelial cells in the abstract and the limitations of use of these cells in interpreting the metabolic consequences of osteocalcin in both the discussion and abstract.

Reviewer #3 (Remarks to the Author):

This revised manuscript addresses most of the comments I raised in my original review. Two points remain to be clarified:

Previous point 2: The Ct value are very low (>30 cycle) for all the qPCR and no information is provided on the amount of RNA used in the synthesis of cDNA. The current data are still not clearly supporting the conclusion that ECs express significant amount of the osteocalcin mRNA. Did they confirmed that their qPCR assays, in particular the ones for osteocalcin, are linear and actually detecting the proper product? Have they performed standard curves and look if a single product was amplified on gel?

Previous point 8: The new IP experiment presented in figure 3B is supposed to show endogenous LRP1 interaction with FOXOs proteins, however the bottom blot is labelled "Flag-Lrp1".

RESPONSE TO THE REVIEWERS

Reviewer #1 (Remarks to the Author):

In the revised manuscript, the authors have addressed all of my concerns. I feel that this is an important study connecting the endothelium, LRP1, osteocalcin and glucose metabolism.

Thank you very much for your comments.

Reviewer #2 (Remarks to the Author):

The authors have answered most questions of reviewers. My only suggestion at this stage is acknowledge the use of lung endothelial cells in the abstract and the limitations of use of these cells in interpreting the metabolic consequences of osteocalcin in both the discussion and abstract.

We appreciate the Reviewer's suggestions. Yes, the use of lung ECs has been acknowledged in the abstract and its limitation has been discussed too.

In the abstract, the following sentence has been updated- "Here we show osteocalcin (OCN), recognized as a bone-secreted metabolic hormone, is expressed in primary endothelial cells (ECs) isolated from mouse heart, lung and liver."

In the discussion, the following sentences has been updated- "Interestingly, OCN mRNA and protein were detected in mouse lung, heart and liver ECs (Fig. 1) but human *Ocn* promoter-driven GFP signals were not observed in these organs (Fig. 1, 2). It suggests the expression pattern of human OCN might be different from mouse OCN. In addition, its expression in mouse tissues other than lung, heart and liver still need further evaluation due to EC heterogeneity. The mouse *Ocn* promoter-driven reporter model would be a great tool for these studies."

Reviewer #3 (Remarks to the Author):

This revised manuscript addresses most of the comments I raised in my original review. Two points remain to be clarified:

Previous point 2: The Ct value are very low (>30 cycle) for all the qPCR and no information is provided on the amount of RNA used in the synthesis of cDNA. The current data are still not clearly supporting the conclusion that ECs express significant amount of the osteocalcin mRNA. Did they confirmed that their qPCR assays, in particular the ones for osteocalcin, are linear and actually detecting the proper product? Have they performed standard curves and look if a single product was amplified on gel?

We totally understand the Reviewer's concerns about qPCR assays. This is how we perform qPCR assays routinely in our laboratory. Before we perform any qPCR assay, PCR efficiency is evaluated for each pair of primers and probes. PCR primer and probe sets with the values at the range 1.90-2.10 will be considered as efficient. Next, we perform cDNA synthesis using 0.5- 1 ug total RNA purified from the primary mouse ECs using iScript cDNA synthesis kit (Bio-Rad, Cat#1708891). These cDNAs are then diluted in 1:2~1:20 for qPCR assays. Our qPCR results present in the manuscript were collected from efficient PCR reactions with validated PCR primers and probes (Universal ProbeLibrary, Roche).

Next, we examined the OCN PCR reaction efficiency with mouse heart and lung endothelial cells (MHLECs). Using the 2-fold serial dilution fractions of MHLEC cDNAs as the templates, we observed a linear relationship between \log_2 (dilution factors) and Ct values between 25~36 and the PCR efficiency was 2.08 (Rebuttal Figure 1a). It suggests that the OCN primers and probe set is efficient for qPCR assays.

To increase the Ct value (<30 cycles) as the Reviewer suggested, we performed additional qPCR experiments with MHLECs using the iScript advanced cDNA synthesis kit (Bio-Rad, Cat#1725038). This kit produces higher yield and could be used with more total RNA (1-7 ug) templates than our previous kit (Bio-Rad, Cat#1708891). In this assay, we synthesized cDNA with 4 ug total RNA, followed by qPCR assays with non-diluted cDNA. The raw readings for OCN were 28.77 or 24.32 in wild-type or LRP1 eKO MLECs, respectively (Supplemental Figure 2e, Rebuttal Table 1). The final products of qPCR reactions were subjected for 2% agarose gel electrophoresis. As expected, a clear single band (92 bp) was detected in each reaction and the difference between LRP1 eKO and wild-type MHLECs was still obvious (Rebuttal Figure 1b).

Rebuttal Table 1. Ct numbers for qPCR assays for OCN in MHLECs.

Ct number	OCN		beta-Actin	
	MHLEC-WT	MHLEC-LRP1 eKO	MHLEC-WT	MHLEC-LRP1 eKO
Mean ± SE	28.77 ± 1.05	24.32 ± 0.92	15.43 ± 0.32	16.28 ± 2.03

In addition, we performed standard curve studies with 10-fold serial dilution fractions of Flag-tagged mouse OCN plasmid DNAs as the templates. The results demonstrated a standard curve with a satisfying PCR efficiency at 1.95 (Supplementary Figure 2f). Based on this standard curve and 1:1 ratio conversion from RNA to cDNA, we calculated OCN RNA copies in LRP1 eKO and wild-type MHLECs. OCN RNA copies were 19-fold higher in LRP1 eKO MHLECs than wild-type cells (Supplementary Figure 2g).

Taken all together, we conclude that OCN mRNA level is upregulated in LRP1-depleted MHLECs.

Previous point 8: The new IP experiment presented in figure 3B is supposed to show endogenous LRP1 interaction with FOXOs proteins, however the bottom blot is labelled "Flag-Lrp1".

We would like to apologize for this error. It should be endogenous LRP1. The label "Flag-Lrp1" has been changed to "LRP1".

Reviewer comments, third round of review:

Reviewer #3 (Remarks to the Author):

The authors have now addressed all my concerns.